# Music Foundation Model as Generic Booster for Music Downstream Tasks

**WeiHsiang Liao**[1*], **Yuhta Takida**[1*], **Yukara Ikemiya**[1*], **Zhi Zhong**[2*],
**Chieh-Hsin Lai**[1], **Giorgio Fabbro**[3], **Kazuki Shimada**[1], **Keisuke Toyama**[2],
**Kin Wai Cheuk**[1], **Marco A. Martínez-Ramírez**[1], **Shusuke Takahashi**[2],
**Stefan Uhlich**[3], **Taketo Akama**[4], **Woosung Choi**[1], **Yuichiro Koyama**[2], **Yuki Mitsufuji**[1,2]

[1] *SonyAI, Tokyo, Japan*
[2] *Sony Group Corporation, Tokyo, Japan*
[3] *Sony Europe B.V., Stuttgart, Germany*
[4] *Sony CSL, Tokyo, Japan*
*Equal contribution

**Reviewed on OpenReview:** *https://openreview.net/forum?id=kHl4JzyNzF*

## Abstract

We demonstrate the efficacy of using intermediate representations from a single foundation model to enhance various music downstream tasks. We introduce SoniDo, a music foundation model (MFM) designed to extract hierarchical features from target music samples. By leveraging hierarchical intermediate features, SoniDo constrains the information granularity, leading to improved performance across various downstream tasks including both understanding and generative tasks. We specifically evaluated this approach on representative tasks such as music tagging, music transcription, music source separation, and music mixing. Our results reveal that the features extracted from foundation models provide valuable enhancements in training downstream task models. This highlights the capability of using features extracted from music foundation models as a booster for downstream tasks. Our approach not only benefits existing task-specific models but also supports music downstream tasks constrained by data scarcity. This paves the way for more effective and accessible music processing solutions.

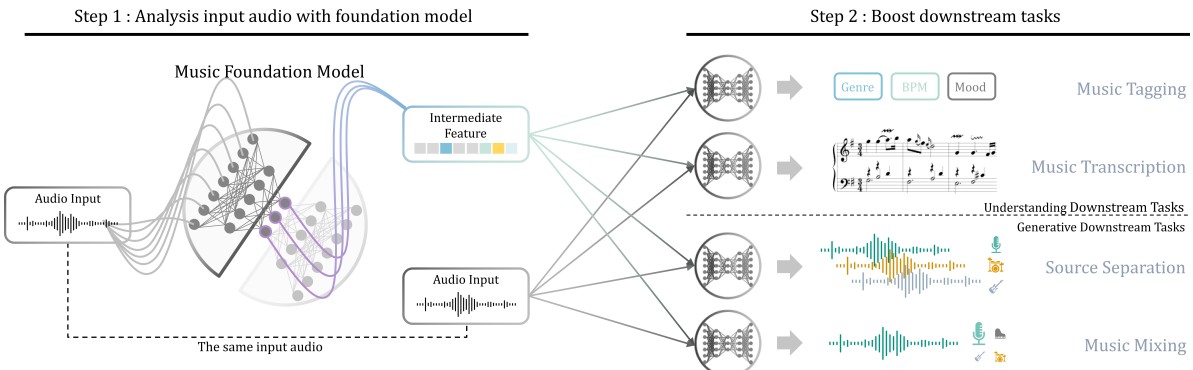

Figure 1: SoniDo extracts hierarchical features of target music samples, which are useful for solving music downstream tasks including understanding and generative tasks.

# 1 Introduction

A foundation model is a pre-trained model developed on a large-scale dataset that can be adapted for a variety of downstream tasks (Bommasani et al., 2021). Several language processing models (Radford et al., 2021; Brown et al., 2020; Devlin et al., 2018; Team et al., 2024) are considered foundation models due to their ability to unify all language tasks as sequence prediction tasks, effectively addressing multiple tasks with a single model. These foundation models have gained significant attraction and are widely used in everyday applications. In contrast, a powerful *music foundation model* capable of handling various *music downstream tasks* for music production is lacking (Ma et al., 2024). We categorize the tasks that a music foundation model primarily addresses into two types: *understanding tasks*, such as tagging and transcription, and *generative tasks*, such as mixing and mastering.

Several multi-task models have been proposed as potential music foundation models (Li et al., 2023a; Yang et al., 2023; Copet et al., 2023; Agostinelli et al., 2023). However, this approach necessitates the inclusion of the desired tasks during the training phase. A notable strategy to overcome this limitation is to inject features extracted from a pre-trained large-scale model into smaller back-end models for downstream tasks that were not seen during training. This ensemble approach, which combines a large-scale model with various smaller models, can effectively function as a music foundation model. The codified audio language modeling (CALM) framework proposed by Castellon et al. (2021) is the first work in this direction, utilizing the intermediate representations from Jukebox (Dhariwal et al., 2020) to tackle music information retrieval (MIR) tasks, covering most music understanding tasks. Beyond MIR,Donahue et al. (2022) leveraged representations from Jukebox for melody transcription. Other studies have followed this approach to address time-invariant MIR tasks using the latest generative models based on residual quantized variational Autoencoders (RQ-VAEs) (Zeghidour et al., 2022; Défossez et al., 2023), enhancing the state-of-the-art (SOTA). However, these applications remain limited to music understanding tasks. Li et al. (2023b) expanded the focus to include music source separation, a generative task, but encountered instability issues during training. The performance of this extension does not yet match that of the baselines mentioned by Mitsufuji et al. (2022). An extensive overview of related work can be found in Appendix A.

We extended the methodology from MIR tasks to generic music downstream tasks. To address both understanding and generative tasks, we focus on a tokenization-based generative modeling approach, which is directly analogous to foundation models in the natural language processing field. Furthermore, We hypothesize that the representation structure of foundation models is crucial in this context. Specifically, we propose that hierarchical representations, which divide information of varying granularity into different levels of embedding, are expected to provide efficient information hierarchy for all downstream tasks including both understanding and generative tasks. We empirically verify this hypothesis in Section 4. In contrast, music foundation models that have been applied to boost music downstream tasks do not have such a hierarchical structure. For example, Jukebox (Castellon et al., 2021; Donahue et al., 2022) is trained to have multi-level representation inspired from hierarchical latent representation (Razavi et al., 2019); however, each level is independently trained. RQ-VAEs (Yang et al., 2023; Li et al., 2023b) learn factorized representation that has a self-organized coarse-to-fine structure, however, they are not hierarchical.

In accordance with the aforementioned hypothesis, we outline this study as follows. We propose and train our music foundation model, SONIDO (meaning *sound* in Spanish), on a high-fidelity internal dataset [1] to establish a task-agnostic feature extraction pipeline. SONIDO is a generative model consisting of a multi-level transformer with a multi-level hierarchical encoder. With proper pre-processing, we infuse its intermediate representation as features to task-specific models on various music downstream tasks with data augmentation. Moreover, for understanding tasks, we proposed an on-the-fly data augmentation called *token-out* to avoid overfitting. Performance evaluation was done by benchmarking with representative tasks from understanding to generative tasks: music tagging, music transcription, music source separation, and music mixing.

---

[1] The rights of this internal dataset are trained on licensed content only. Except for as specifically authorized by the rights owner, the rights owner expressly prohibits and has opted out of any text or data mining, web scraping or similar, reproductions, extractions or uses, of its content for any purposes, including in relation to training, developing, or commercializing any AI System.

Table 1: Performance overview of applying extracted features to various downstream tasks. Bold: best, underline: second best. The result marked with * is obtained with a different evaluation protocol. The results marked with † are numbers reported in Castellon et al. (2021).

| Downstream Task | Dataset | Metric | SoniDo | MusicGen Small | MusicGen Large | Jukebox-5B | MERT | Task-Specific SOTA |
|---|---|---|---|---|---|---|---|---|
| Multi-task Music Tagging | MusicTagATune | ROC-AUC | 91.7 | 90.4 | 90.5 | 91.5† | 91.3 | **92.0** | |
| | | mAP | **41.5** | 38.8 | 39.0 | 41.4† | 40.2 | 38.4 | (Huang et al., 2022a) |
| Pitch Estimation | Nsynth | Acc. | 93.8 | 93.3 | 92.8 | 91.6† | **94.4** | 89.2 | (McCallum et al., 2022) |
| Instrument Classification | | Acc. | 78.0 | 71.9 | 74.2 | 70.4† | 72.6 | **78.2** | (Wang et al., 2022) |
| Emotion Regression | EmoMusic | Averaged $R^2$ | 64.7 | 45.6 | 46.2 | 66.9† | **68.0** | 63.0* | (Castellon et al., 2021) |
| Key Detection | GiantSteps | Weighted Acc. | 63.5 | 65.2 | 62.4 | 66.7† | 65.6 | **79.6** | (Castellon et al., 2021) |
| Genre Classification | GTZAN | Acc. | 80.7 | 75.2 | 70.3 | 79.7† | 79.3 | **83.5** | (McCallum et al., 2022) |
| Singer Identification | VocalSet | Acc. | 87.0 | 82.3 | 83.3 | 82.6† | **87.1** | 80.3 | (Modrzejewski et al., 2023) |
| Technique Identification | | Acc. | 74.4 | 66.1 | 63.9 | 76.7† | **76.9** | 65.6 | (Yamamoto et al., 2022) |
| Music Transcription | MAPS | Frame F1 | 83.92 | 82.94 | 81.53 | **84.23** | - | 82.89 | |
| | | Note F1 | 86.45 | 85.97 | 85.14 | **86.54** | - | 85.14 | |
| | | Note w/ Offset F1 | 68.27 | **68.27** | 66.28 | 68.26 | - | 66.34 | (Toyama et al., 2023) |
| | | Note w/ Offset & Velocity F1 | **51.34** | 50.42 | 48.69 | 50.46 | - | 48.20 | |
| Source Separation | MUSDB18 | SDR (bass) | 9.50 | 8.86 | 8.17 | 7.12 | 5.6 | **11.31** | |
| | | SDR (drums) | 8.65 | 8.03 | 7.50 | 6.65 | 3.6 | **9.49** | |
| | | SDR (other) | 5.91 | 5.59 | 5.54 | 4.77 | 3.0 | **7.73** | (Lu et al., 2024) |
| | | SDR (vocals) | 8.07 | 7.57 | 7.66 | 6.84 | 5.3 | **10.66** | |
| | MDXDB21 hidden | SDR (bass) | **8.14** | 7.44 | 7.40 | 6.58 | - | 7.86 | |
| | | SDR (drums) | 8.16 | **8.31** | 7.37 | 6.58 | - | 7.89 | |
| | | SDR (other) | 5.21 | **5.26** | 4.93 | 4.59 | - | 5.09 | (Rouard et al., 2023) |
| | | SDR (vocals) | **8.04** | 7.81 | 7.73 | 7.12 | - | 7.70 | |
| Music Mixing | MDXDB21-dry hidden | Stereo-Invariant | **79.86** | 87.27 | 87.32 | 87.97 | - | 82.09 | |
| | | $\text{Spectral}_{mape}$ | 0.221 | 0.229 | 0.228 | 0.231 | - | **0.193** | |
| | | $\text{Panning}_{mape}$ | **0.175** | 0.244 | 0.219 | 0.249 | - | 0.179 | (Martínez-Ramírez et al., 2022) |
| | | $\text{Dynamic}_{mape}$ | **0.064** | 0.072 | 0.073 | 0.075 | - | 0.070 | |
| | | $\text{Loudness}_{mape}$ | 0.171 | 0.148 | **0.132** | 0.144 | - | 0.152 | |

The encoder design of SoniDo is inspired by Jukebox but makes the representation hierarchical by enforcing the fine level to be conditioned by the coarse levels using a hierarchical autoencoder framework called hierarchically quantized VAE (HQ-VAE) (Takida et al., 2024). We then use a transformer-based multi-level auto-regressive model to characterize the probability mass of learnt HQ-VAE embeddings. We extract features from the intermediate representation of SoniDo by first converting input audio with the encoder into tokens, feeding them into the transformers, and extracting the intermediate output from the midst layer. We refer to these extracted features as SoniDo features.

As shown in Table 1, we test SoniDo's feature injection for selected downstream tasks along with several baselines. To the best of our knowledge, this is the first study on enhancing both understanding and generative tasks with the intermediate representation from a single model. We briefly list our major findings:

1. We empirically show that, with an auto-regressive generative model that is established on hierarchical representation, its intermediate representation can serve as generic booster of various music downstream tasks.

2. We verify that the extracted intermediate representation is beneficial for music understanding tasks even with only an extra shallow back-end network. The extension of the shallow network with attention layers leads to further improvement.

3. We show that the extracted intermediate representation is beneficial for enhancing task-specific models, through the applications to both understanding and generative tasks.

4. Several of the above improvements in each task category result in new SOTA scores. The summary of our results is shown in Table 1.

## 2 Proposed Two-stage Hierarchical Model: SoniDo

To explore the effectiveness of hierarchical modeling in boosting downstream tasks, we adopt a typical two-stage generative modeling (Dhariwal et al., 2020; Copet et al., 2023; Li et al., 2023a). In stage-1, we use an HQ-VAE for hierarchical representation learning, dividing information into different levels based on granularity. In stage-2, we use auto-regressive modeling to learn the multi-level token streams extracted from the stage-1 model. Finally, features from stage-2 model are extracted as described in Section 3.1.

## 2.1 Stage-1 Model: HQ-VAE

We construct the architecture of SoNiDo to learn a hierarchical representation of the target dataset. Consider a music sample $\boldsymbol{x}$ with length $T$, where $\boldsymbol{x} \in \mathcal{X} \subset \mathbb{R}^T$. A set of codebooks $\{\boldsymbol{B}_1, \boldsymbol{B}_2, \boldsymbol{B}_3\}$ is used for learning a three-layer hierarchical representation on $\boldsymbol{x}$. For $l \in \{1, 2, 3\}$, the $l$th codebook is denoted as $\boldsymbol{B}_l = \{\boldsymbol{b}_{l,k}\}_{k=1}^{K_l}$, which consists of $K_l$ $d_l$-dimensional trainable vectors $\boldsymbol{b}_{l,k} \in \mathbb{R}^{d_l}$. The architecture is designed to extract a hierarchical latent representation of music samples, which is denoted as $\boldsymbol{Z}_{1,2,3} := \boldsymbol{Z}_1 \otimes \boldsymbol{Z}_2 \otimes \boldsymbol{Z}_3$ with $\boldsymbol{Z}_l \in \boldsymbol{B}_l^{t_l}$ $(l = 1, 2, 3)$, where $t_l$ is the latent sequence length at the $l$th layer. The discrete tensors $\boldsymbol{Z}_1$, $\boldsymbol{Z}_2$, and $\boldsymbol{Z}_3$ are expected to convey the coarse, medium, and fine-grained information. The reconstruction can be done with a well-optimized neural function $\boldsymbol{f} : \boldsymbol{B}_1^{t_1} \otimes \boldsymbol{B}_2^{t_2} \otimes \boldsymbol{B}_3^{t_3} \to \mathcal{X}$, i.e., $\boldsymbol{x} \approx \boldsymbol{f}(\boldsymbol{Z}_{1,2,3})$.

The architecture is composed of bottom-up and top-down paths, as illustrated in Figure 2(a), the inference process of which is as follows. A series of encoders in the bottom-up path extracts feature tensors for three different information resolutions, which are denoted as $\boldsymbol{H}_l(\boldsymbol{x})$ $(l = 1, 2, 3)$, from sample $\boldsymbol{x}$. The feature $\boldsymbol{H}_l(\boldsymbol{x})$ is used for the top-down path to process the data in a hierarchical manner. The top-down path has three (top, middle, and bottom) top-down blocks to model hierarchical discrete latent representations. The top block first quantizes $\tilde{\boldsymbol{Z}}_1 := \boldsymbol{H}_1(\boldsymbol{x})$, which has the most global (coarse) information amongst the encoded features, into discrete tensor $\boldsymbol{Z}_1$ by the nearest neighbor search in codebook $\boldsymbol{B}_1$. At the next step, the middle latent tensor is conditioned on the top $\boldsymbol{Z}_1$ to focus more on local details, with the injection of $\boldsymbol{H}_2(\boldsymbol{x})$. Therefore, the block takes both tensors processed in the top block and bottom-up paths, i.e., $\boldsymbol{Z}_1$ and $\boldsymbol{H}_2(\boldsymbol{x})$, generating a raw continuous feature $\tilde{\boldsymbol{Z}}_2 := \boldsymbol{G}_2(\boldsymbol{H}_2(\boldsymbol{x}), \boldsymbol{Z}_1)$. The raw feature is then quantized into $\boldsymbol{Z}_2$ in the same manner as with codebook $\boldsymbol{B}_2$. The bottom block repeats a similar process with $\boldsymbol{Z}_2$ and $\boldsymbol{H}_3(\boldsymbol{x})$ to further refine the representation with the additional discrete feature $\boldsymbol{Z}_3$. Finally, the set of $\boldsymbol{Z}_1$, $\boldsymbol{Z}_2$, and $\boldsymbol{Z}_3$ is decoded to the data space to reconstruct $\boldsymbol{x}$.

We train the architecture including the codebooks within the variational Bayes framework, as an instance of HQ-VAE, stochastically quantized VAE-2 (SQ-VAE-2) (Takida et al., 2024). To establish a generative process in this VAE, we first define the prior probability distribution on $\boldsymbol{Z}_{1,2,3}$ as $P(\boldsymbol{Z}_{1,2,3}) = P_1(\boldsymbol{Z}_1)P_2(\boldsymbol{Z}_2|\boldsymbol{Z}_1)P_3(\boldsymbol{Z}_3|\boldsymbol{Z}_{1,2})$. Given a chunk of latent variables $\boldsymbol{Z}_1$, $\boldsymbol{Z}_2$, and $\boldsymbol{Z}_3$, a data sample can be generated under a conditional probability distribution $p(\boldsymbol{x}|\boldsymbol{Z}_{1,2,3})$. Concretely, we parameterize the conditional distribution as a normal distribution with function $\boldsymbol{f}$ and a trainable isotropic covariance matrix as $p(\boldsymbol{x}|\boldsymbol{Z}_{1,2,3}) = \mathcal{N}(\boldsymbol{f}(\boldsymbol{Z}_{1,2,3}), \sigma^2 \boldsymbol{I})$. To summarize, the generative process consists of two steps: sampling $\boldsymbol{Z}_{1,2,3}$ from the prior distribution and decoding it with the conditional distribution. Note that, in practice, $\boldsymbol{Z}_{1,2,3}$ is sampled from an estimated posterior distribution instead of the prior distribution, as presented in Section 2.2. Next, the approximated posterior distribution for $p(\boldsymbol{Z}_{1,2,3}|\boldsymbol{x})$ is set as $Q(\boldsymbol{Z}_{1,2,3}|\boldsymbol{x}) = Q_1(\boldsymbol{Z}_1|\boldsymbol{x})Q_2(\boldsymbol{Z}_2|\boldsymbol{Z}_1, \boldsymbol{x})Q_3(\boldsymbol{Z}_3|\boldsymbol{Z}_{1,2}, \boldsymbol{x})$. We connect each $Q_1$, $Q_2$, and $Q_3$ with the components in Figure 2(a). Specifically, the categorical distribution at the $l$th layer, $Q_l(\boldsymbol{Z}_l|\boldsymbol{Z}_{<l}, \boldsymbol{x})$, is defined as a stochastic quantization that is $\hat{P}_{s_l^2}(\boldsymbol{z}_{l,n} = \boldsymbol{b}_{l,k}|\tilde{\boldsymbol{z}}_{l,n}) \propto \exp(-\|\tilde{\boldsymbol{z}}_{l,n} - \boldsymbol{b}_{l,k}\|^2/2s_l^2)$ with a trainable positive scalar $s_l^2$, where $\boldsymbol{z}_{l,n}$ and $\tilde{\boldsymbol{z}}_{l,n}$ indicate the $n$th vectors in $\boldsymbol{Z}_l$ and $\tilde{\boldsymbol{Z}}_l$, respectively. Finally, the resulting training objective consists of terms for reconstruction and latent regularization:

$$\mathcal{L}_1(\boldsymbol{x}) = \frac{T}{2} \log \sigma^2 + \mathbb{E}_{Q(\boldsymbol{Z}_{1,2,3}|\boldsymbol{x})} \left[ \frac{\|\boldsymbol{x} - \boldsymbol{f}(\boldsymbol{Z}_{1,2,3})\|_2^2}{2\sigma^2} + \sum_{l=1}^{3} \left( \frac{\|\tilde{\boldsymbol{Z}}_l - \boldsymbol{Z}_l\|_F^2}{2s_l^2} - H(\hat{P}_{s_l^2}(\boldsymbol{Z}_l|\tilde{\boldsymbol{Z}}_l)) \right) \right], \quad (1)$$

where $H(\cdot)$ is the entropy of a probability mass function. Progressive coding (Takida et al., 2024) is applied to ensure the amount of information is balanced across the three layers.

## 2.2 Stage-2 Model: Sparse Transformers

The stage-2 model addresses the gap between the pre-set prior distribution (i.e., $P(\boldsymbol{Z}_{1,2,3})$) and marginalized posterior distribution (i.e., $Q(\boldsymbol{Z}_{1,2,3}) := \mathbb{E}_{p(\boldsymbol{x})}[Q(\boldsymbol{Z}_{1,2,3}|\boldsymbol{x})]$) by directly learning the posterior distribution. We incorporated the contrastive language-audio pretraining (CLAP) model proposed by LAION (Wu* et al., 2023) into the stage-2 model. To include the CLAP conditioning, we approximate $Q_\phi(\boldsymbol{Z}_{1,2,3})$ with a conditioned decomposition as

$$P_{\boldsymbol{\Pi}}(\boldsymbol{Z}_{1,2,3}|\boldsymbol{y}_{\text{audio}}) = P_{\boldsymbol{\pi}_1}(\boldsymbol{Z}_1|\boldsymbol{y}_{\text{audio}})P_{\boldsymbol{\pi}_2}(\boldsymbol{Z}_2|\boldsymbol{Z}_1, \boldsymbol{y}_{\text{audio}})P_{\boldsymbol{\pi}_3}(\boldsymbol{Z}_3|\boldsymbol{Z}_{1,2}, \boldsymbol{y}_{\text{audio}}), \quad (2)$$

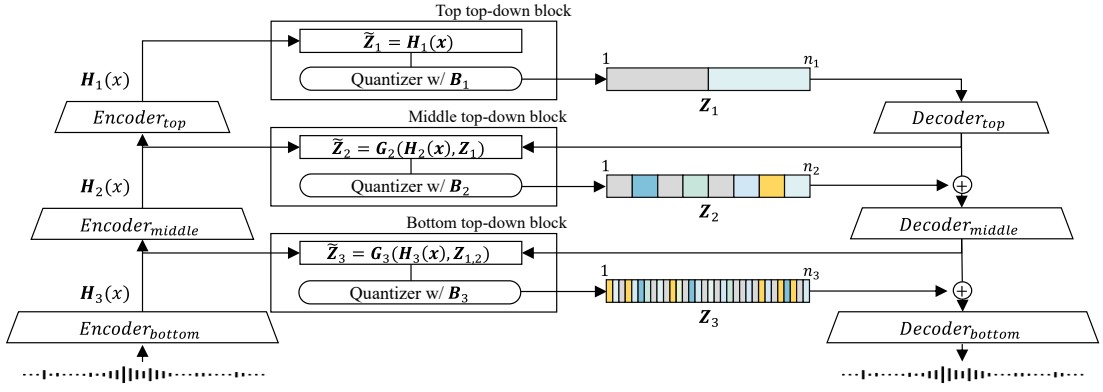

(a) Stage 1 model architecture

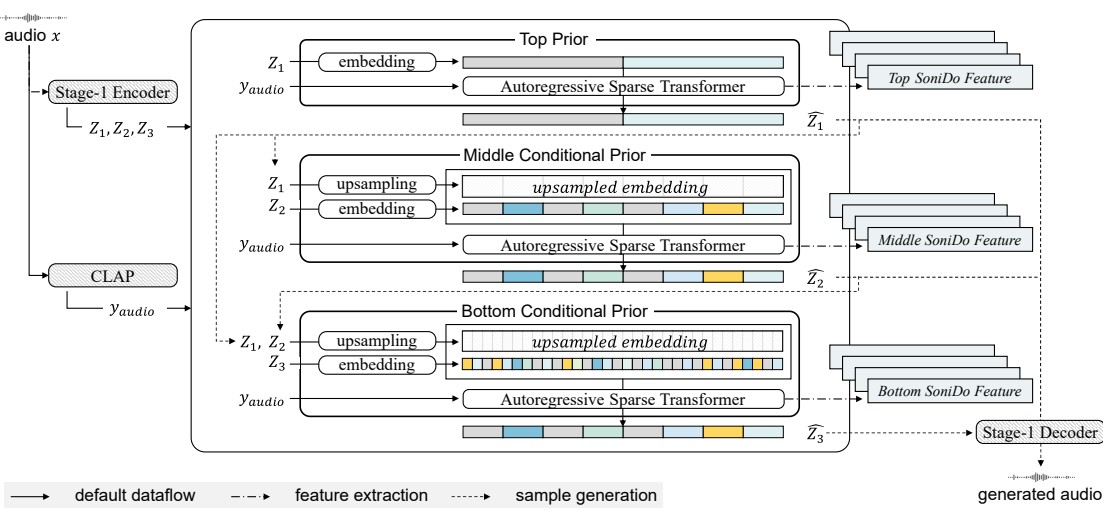

(b) Stage-2 model architecture

Figure 2: The two stages of SONIDO.

where $\boldsymbol{\Pi} := \{\boldsymbol{\pi}_1, \boldsymbol{\pi}_2, \boldsymbol{\pi}_3\}$ is a set of neural networks for the stage-2 model, and $\boldsymbol{y}_{\text{audio}} \in \mathbb{R}^{512}$ denotes the feature produced from the CLAP encoder. Thanks to the alignment between the audio and text embeddings of CLAP, even if the audio dataset has no text caption, we can still feed audio in the training phase, whereas it allows either audio or text input in the inference stage. The use of a pre-trained encoder is common in modern generative models. For example, MusicGen (Copet et al., 2023) uses the pre-trained T5 encoder (Raffel et al., 2019) to model the text conditions. The training objective is negative log-likelihood:

$$\mathcal{L}_2(\boldsymbol{x}) = \mathbb{E}_{Q_{\boldsymbol{\phi}}(\boldsymbol{Z}_{1,2,3}|\boldsymbol{x})p_{\text{CLAP}}(\boldsymbol{y}_{\text{audio}}|\boldsymbol{x})}[-\log P_{\boldsymbol{\Pi}}(\boldsymbol{Z}_{1,2,3}|\boldsymbol{y}_{\text{audio}})]. \qquad (3)$$

We follow Jukebox (Dhariwal et al., 2020) to construct the networks $\boldsymbol{\Pi}$ with sparse transformers (Vaswani et al., 2017; Child et al., 2019). As illustrated in Figure 2(b), we train three auto-regressive sparse transformers to model $P(\boldsymbol{Z}_1|\boldsymbol{y}_{\text{audio}})$, $P(\boldsymbol{Z}_2|\boldsymbol{Z}_1, \boldsymbol{y}_{\text{audio}})$, and $P(\boldsymbol{Z}_3|\boldsymbol{Z}_{1,2}, \boldsymbol{y}_{\text{audio}})$, which we refer to as top prior, middle conditional prior, and bottom conditional prior, respectively. The middle and bottom priors use the token sequences from the upper levels, with up-sampling achieved through *upsampling modules*, corresponding to the *conditioners* of Jukebox. We additionally condition each prior on $\boldsymbol{y}_{\text{audio}}$. Appendix B.2 provides further details of the stage-2 model. Appendix B.3 evaluates the common objective metrics on SONIDO.

### 2.3 SONIDO **vs. Other Music Foundation Models**

This section compares the architecture of SONIDO with those of other well-known music foundation models, i.e., Jukebox (Dhariwal et al., 2020), MusicLM (Agostinelli et al., 2023), and MusicGen (Copet et al., 2023). These models are categorized on the basis of how their stage-1 models are constructed; (a) SQ-VAE-2, (b) multi-resolution VQ-VAEs, and (c) residual vector quantization (RVQ), as illustrated in Figure 3.

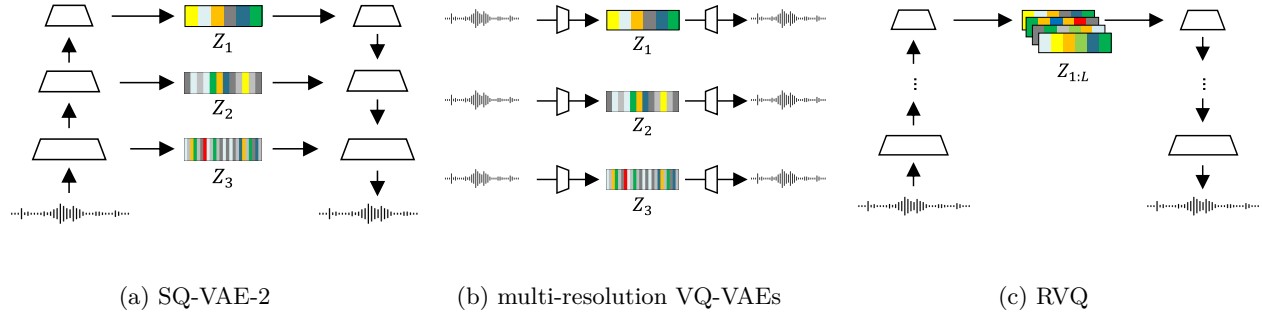

(a) SQ-VAE-2       (b) multi-resolution VQ-VAEs       (c) RVQ

Figure 3: Stage-1 model comparison.

While SONIDO and Jukebox exhibit some shared characteristics, such as a three-level architecture in the stage-1 model, SONIDO is based on SQ-VAE-2, whereas Jukebox used multi-resolution VQ-VAEs. In Jukebox, token streams $Z_1$, $Z_2$, and $Z_3$ were independently and separately trained for different sampling rates. Consequently, the $l$th transformer was designed to generate $Z_l$ by *upsampling* the previous token sequence $Z_{l-1}$ for $l = 2, 3$. In contrast, SONIDO's token streams from the stage-1 model are jointly trained and collaboratively contribute to the comprehensive modeling of the waveform at the original sampling rate from scratch. Given the tight interrelation between token streams from different levels, SONIDO's $l$th transformer is conditioned on all the upper token streams.

Recent approaches such as MusicLM and MusicGen used RVQ in a bottleneck feature space instead of applying these hierarchical quantization methods (e.g., SQ-VAE-2). These approaches also use transformers to model the prior of the music-token streams $P(Z_{1:L})$. In the context of token-sequence length for generating 1 s of audio at a target sampling rate $sr$, the bottom-most token-sequence length in SONIDO and Jukebox is $sr/8$, while MusicGen requires a token-sequence length of $sr/640$. RVQ-based models excute quantization in highly compressed latent spaces using a series of vector quantization layers, effectively shortening the token sequence to be learned by transformers in the stage-2 model.

## 3 SONIDO **on Music Downstream Tasks**

We first obtain SONIDO features from input audio with a task-agnostic feature extraction process. Depending on whether the downstream task is time-invariant or time-varying, we then apply different pre-processing steps. Finally, we inject the pre-processed SONIDO features into a proper location of a target task-specific model. The selection of such a location is explained in Section 4.

### 3.1 Task-agnostic Feature Extraction

We follow the feature extraction pipeline in Castellon et al. (2021); Niizumi et al. (2022); Huang et al. (2022b) based on the pre-trained frozen SONIDO. The music waveform is first converted to multi-level token sequences via the stage-1 encoder of SONIDO. The tokens are then fed into the top prior, middle conditional, and bottom conditional priors of SONIDO without auto-regressive iteration. The middle and bottom priors are conditioned by the ground-truth tokens produced with the stage-1 encoders. We extract the output of the $N$-th ($N = 36$ as in Castellon et al. (2021)) transformer layer in those prior models as SONIDO features. If the CLAP audio embedding is not used as the condition of priors, we call the feature extraction unconditional extraction; otherwise, CLAP-conditional extraction.

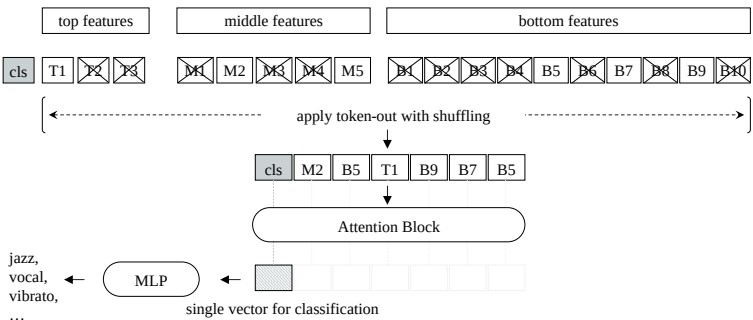

Figure 4: Attention-based feature aggregation and token-out data augmentation. "T", "M", "B" mean top, middle, and bottom priors, respectively. Token-out augmentation deletes masked tokens from input sequence. Attention block aggregates sequence into single vector and is followed by MLP to predict tags.

The maximum sequence length of the priors is 8192. Since the down-sampling rates in the stage-1 model are $128\times$ (top), $32\times$ (middle), and $8\times$ (bottom), the same amount of 8192 tokens in different priors correspond to 24 s, 6 s and 1.5 s in the time domain, forming a set of hierarchical multi-rate features. To save computational resources, the SONIDO features are pre-computed for most downstream tasks, except for HTDemucs mentioned in Section 4.2.2, where a clip of music is randomly selected on-the-fly during training. To compute features for a long audio input, we treat the input as overlapping segments with the ratio $N_{\mathrm{ovlp}}$. If $N_{\mathrm{ovlp}}$ is sufficiently large such that the overlap is longer than the perception field of the stage-2 model, it is guaranteed that the feature extraction result is not affected by the segmentation.

## 3.2 Feature Pre-processing for Time-invariant Downstream Tasks

If the downstream task is time-invariant, we first divide input audio into non-overlapping segments of 24 (top), 6 (middle), and 1.5 (bottom) s. For each prior, the SONIDO features of the segment are reduced to a single token via average pooling, forming 3 SONIDO token sequences in the end.

The common practice (Castellon et al., 2021; Li et al., 2023b) suggests using a multi-layer perceptron (MLP) with a single hidden layer of 512 dimensions to probe the features. However, SONIDO token sequences originate from priors with different time resolutions, which is different from prior studies. To effectively use these hierarchical features, a sequence aggregation is required. We thus propose to aggregate the sequences via a standard attention block, which is an attention layer followed by a feed-forward layer. This is inspired by the attention-based feature aggregation in instrument classification tasks (Gururani et al., 2019; Zhong et al., 2023a). We first concatenate the hierarchical SONIDO features into a single token sequence then attach a learnable class token at the front. The attention block is trained to aggregate all features into the class token, which is then converted to music tags or emotion scores by the aforementioned MLP. Hyperparameters as well as an ablation study on the sequence aggregation are provided in Appendix C.

To prevent overfitting when using the concatenated token sequence to train the attention block, we propose an on-the-fly data augmentation method called *token-out*. This is inspired by SpecAugment (Park et al., 2019) and masked Transformers (Koutini et al., 2022; Zhong et al., 2023b; Comunità et al., 2024), in which a part of the input is masked before feeding into deep neural networks. Unlike prior arts, token-out is applied to the whole *token* sequence extracted with the multiple layers in SONIDO, as illustrated in Figure 4. The masking ratio is sampled between 0 and 100% uniformly. As shown in Appendix C, aggregating the SONIDO features with the shallow attention layer and token-out augmentation led to performance improvement.

## 3.3 Feature Pre-processing for Time-varying Downstream Tasks

When applying the SONIDO features to a task-specific model on a time-varying downstream task, we face several challenges, such as temporal alignment, proper amount of information compression, and sufficient feature adaptation before injecting features into the task-specific models.

The temporal alignment between the SONIDO features and target model can be achieved by either pooling the SONIDO features or using linear layers. Examples of these two cases are provided in Appendices E.2 and D, respectively. Compared with average pooling or max-pooling, we found that using linear layers can yield better performance, as described in Appendix E.2.

Both information compression and feature adaptation can be done with linear layers. The output dimension is simply set to match the feature dimension of the target task-specific model. Empirically, one layer is sufficient for most of the models we have tested, except for music transcription with hFT-Transformer Toyama et al. (2023), which requires four layers, as described in Appendix D.3.

## 4 Experiments

We conducted experiments to examine the usefulness of features extracted from music foundation models for understanding and generative downstream tasks by addressing two questions: **Q1:** Do extracted features have useful information for music understanding? **Q2:** Can extracted features boost current task-specific models for both understanding and generative tasks? To verify the generalizability of the results, we test not only the SONIDO features, but also features extracted from Jukebox and the two public versions of MUSICGEN (Copet et al., 2023), namely MUSICGEN SMALL and MUSICGEN LARGE. As a major focus of this work is to extend the applicable downstream tasks, we evaluate Jukebox's features specifically for time-varying tasks and report the results from Castellon et al. (2021) for time-invariant tasks in Table 1.

To address the first question, we selected eight music tagging tasks and music transcription as representatives for understanding tasks.We verified that the extracted features encompass both time-invariant information of overall musical properties and time-varying information of specific musical events. For the second question, we tested the injection of extracted features into several task-specific models for understanding and generative tasks. They consist of one transcription model (Toyama et al., 2023), two separation models (Mitsufuji et al., 2022; Fabbro et al., 2023), and two mixing models from Martínez-Ramírez et al. (2022).

The feature extraction described in Section 3.1 is used for all experiments. Before applying the features into downstream tasks, task-dependent pre-processing is applied (see Sections 3.2 and 3.3). Details of the experimental setup, results, and further ablation studies are provided in Appendices C, D, E, and F, respectively. We only use features extracted from the top and middle layers of SONIDO. In the preliminary experiments, we found that including the bottom-layer features does not always improve the performance of understanding tasks. We assume this is due to the bottom layer mostly containing only the fine-grained information irrelevant to the tasks, thus degrading performance. This is discussed further in each subsection.

### 4.1 Usefulness of Extracted Features for Music Understanding

The following feature probing experiments demonstrate that features extracted from SONIDO, MUSICGEN and Jukebox all contain valuable information, which is consistent with Castellon et al. (2021). However, for music transcription, the shallow network remains insufficient to match SOTA models. In Section 4.2, we show that injecting extracted features into task-specific models can boost their performance beyond SOTA.

### 4.1.1 Music Tagging

We test a wide range of music tagging tasks as well as the emotion regression task: MagnaTagATune (MTAT) (Law et al., 2009) for auto tagging, Nsynth (Engel et al., 2017) for pitch and instrument recognition, EmoMusic (Soleymani et al., 2013) for emotion regression, GTZAN (Tzanetakis & Cook, 2002) for genre classification, GiantSteps (Knees et al., 2015; Korzeniowski & Widmer, 2017) for musical key estimation, and VocalSet (Wilkins et al., 2018) for singer and singing technique identification. A summary of the datasets is shown in Table 9 in Appendix C. We followed the pre-processing in previous studies (Li et al., 2023b; Yuan et al., 2023) and used scikit-learn (Pedregosa et al., 2011) and mir_eval (Raffel et al., 2014) for metric computation. The average $R^2$ of arousal and valence axis is reported for EmoMusic. The feature pre-processing for time-invariant downstream tasks in Section 3.2 including the feature aggregation and token-out augmentation is applied for all tasks. Following common practice (Castellon et al., 2021; Li et al., 2023b), an MLP is then used to probe the aggregated features.

Table 2: Music tagging. Benchmark results of SoniDo features in music tagging tasks (**bold**: top-2 score).

| Dataset Task | MTAT Auto tagging | | Nsynth Pitch | Nsynth Instrument | EmoMusic Emotion regression | GiantSteps Key | GTZAN Genre | VocalSet Singer | VocalSet Vocal techniques |
|---|---|---|---|---|---|---|---|---|---|
| Metrics | ROC-AUC | mAP | Acc. | Acc. | Average $R^2$ | Weighted acc. | Acc. | Acc. | Acc. |
| *Supervised* | | | | | | | | | |
| MusiCNN Pons & Serra (2019) | 90.6 | 38.3 | 64.1 | 72.6 | 58.5 | 12.8 | 79.0 | 57.0 | 70.3 |
| MULE-supervised McCallum et al. (2022) | **91.7** | 41.3 | 79.3 | 73.1 | 64.6 | 28.6 | **83.5** | - | - |
| *Auto-regression* | | | | | | | | | |
| Jukebox Dhariwal et al. (2020); Castellon et al. (2021) | 91.5 | **41.4** | 91.6 | 70.4 | **66.9** | **66.7** | 79.7 | 82.6 | **76.7** |
| MusicGen-small Copet et al. (2023) | 90.4 | 38.8 | 93.3 | 71.9 | 45.6 | 65.2 | 75.2 | 82.3 | 66.1 |
| MusicGen-large Copet et al. (2023) | 90.5 | 39.0 | 92.8 | 74.2 | 46.2 | 62.4 | 70.3 | 83.3 | 63.9 |
| *Contrastive* | | | | | | | | | |
| CLMR Spijkervet & Burgoyne (2021) | 89.4 | 36.1 | 47.0 | 67.9 | 56.8 | 14.9 | 68.6 | 49.9 | 58.1 |
| Slowfast-NFNet-F0 Wang et al. (2022) | - | 39.5 | 88.0 | **78.2** | - | - | - | - | - |
| MULE-contrastive McCallum et al. (2022) | 91.4 | 40.4 | 89.2 | 74.0 | 63.9 | **66.7** | 73.5 | **87.5** | 75.5 |
| *Mask reconstruction* | | | | | | | | | |
| HuBERT music Hsu et al. (2021); Li et al. (2023b) | 90.2 | 37.7 | 77.4 | 69.3 | 54.3 | 14.7 | 70.0 | 75.3 | 65.9 |
| data2vec music Baevski et al. (2022); Li et al. (2023b) | 90.0 | 36.2 | 93.1 | 69.4 | 61.6 | 50.6 | 74.1 | 81.4 | 71.1 |
| MERT-330M Li et al. (2023b) | 91.3 | 40.2 | **94.4** | 72.6 | **68.0** | 65.6 | 79.3 | **87.1** | **76.9** |
| *Hierarchical auto-regression (ours)* | | | | | | | | | |
| SoniDo | **91.7** | 41.5 | **93.8** | **78.0** | 64.7 | 63.5 | **80.7** | 87.0 | 74.4 |

Table 3: Results of feature probing using shallow back-end on MAPS (**bold**: best, underline: second-best).

| Input | Note F1(%) |
|---|---|
| Spectrogram | 18.83 |
| Spectrogram + SoniDo Top | 57.20 |
| Spectrogram + SoniDo Middle | 64.98 |
| Spectrogram + SoniDo Top + SoniDo Middle | **66.02** |
| Spectrogram + MusicGen Small | 53.18 |
| Spectrogram + MusicGen Large | 49.16 |
| Spectrogram + Jukebox | 57.13 |

We conduct a preliminery study for SoniDo with top prior features to compare CLAP-conditional extraction, unconditional extraction and features from the CLAP encoder. We use the tagging task for coarse-grained concepts on MTAT, and the classification task for fine-grained concepts (pitch) on Nsynth. We found that CLAP performs well for coarse concepts, while unconditional extraction results in better accuracy for pitch estimation. CLAP-conditional extraction achieves better scores in both tasks. Details can be found in Appendix C and Table 10. Consequently, we report SoniDo's scores with the CLAP-conditional feature extraction for time-invariant understanding tasks.

The test results on various datasets and benchmarks with prior studies are listed in Table 2. Probing the SoniDo and MusicGen features both shown competitive scores in most tasks. The SoniDo's features reached the top-2 performance in auto tagging, pitch estimation, instrument classification, and genre classification. They also performs well for emotion regression and singer identification. While these are prompt-conditioned generative models, feature probing using these models reached comparable performance compared with SOTA encoder-only models specialized for understanding tasks.

### 4.1.2 Music Transcription

Beyond time-invariant understanding tasks, we continue the test on music transcription, which is a time-varying understanding task. All of SoniDo, MusicGen, and Jukebox features are obtained with unconditional extraction described in Section 3.1. The dimension and time resolution of extracted features are aligned to those of the spectrogram with linear layers. Following Castellon et al. (2021), the features are concatenated with the spectrogram. A single-layer shallow back-end network is used to probe these features.

We show the transcription performance of the feature probing mentioned above on the MAPS dataset (Emiya et al., 2010) in Table 3. All the features greatly improved the note-wise F1 score compared with using the spectrogram only. This suggests that all of SoniDo, MusicGen, and Jukebox features contain useful information for time-varying understanding tasks.

Table 4: Music transcription results of F1 scores on MAPS (**bold**: best score, underline: second best). "Note" refers to note-wise estimation. First row corresponds to hFT-Transformer (Toyama et al., 2023).

| Training data | Input | Frame | Note | Note w/ Offset | Note w/ Offset&Velocity |
|---|---|---|---|---|---|
| 100[%] | Spectrogram | 82.89 | 85.14 | 66.34 | 48.20 |
| | Spectrogram + SONIDO Top | 83.92 | 86.45 | 68.27 | **51.34** |
| | Spectrogram + SONIDO Top + SONIDO Middle | 84.16 | 85.96 | 67.37 | 50.98 |
| | Spectrogram + MUSICGEN SMALL | 82.94 | 85.97 | **68.27** | 50.42 |
| | Spectrogram + MUSICGEN LARGE | 81.53 | 85.14 | 66.28 | 48.69 |
| | Spectrogram + Jukebox | **84.23** | **86.54** | 68.26 | 50.46 |
| 10[%] | Spectrogram | 9.83 | 0.59 | 0.17 | 0.46 |
| | Spectrogram + SONIDO Top | 65.91 | 66.64 | 39.88 | 25.87 |
| | Spectrogram + SONIDO Top + SONIDO Middle | **71.57** | **75.00** | **46.18** | **30.63** |
| | Spectrogram + MUSICGEN SMALL | 63.73 | 65.90 | 39.00 | 24.94 |
| | Spectrogram + MUSICGEN LARGE | 61.81 | 63.27 | 37.03 | 24.01 |
| | Spectrogram + Jukebox | 70.43 | 73.76 | 45.80 | 30.42 |

## 4.2 Using Extracted Features to Boost Existing Task-specific Models

In Section 4.1, we showed that the extracted features contain useful knowledge for music understanding. In this section, we test all of SONIDO, MUSICGEN, and Jukebox features on several SOTA task-specific models, the tasks include music transcription, music source separation, and music mixing, which covered both music understanding and generative tasks. The experimental results indicate that the extracted features consistently boost the performances of task-specific models. We also observed that injecting the SONIDO features accelerated the decrement of training loss in early epochs.

### 4.2.1 Music Transcription: hFT-Transformer

We applied the extracted features to hFT-Transformer (Toyama et al., 2023), a SOTA music transcription model for piano on MAPS (Emiya et al., 2010), to assess whether it surpasses existing models that rely solely on the spectrogram. On the basis of the input spectrogram, hFT-Transformer estimates the frame-based note activation, along with the onset, offset, and velocity of a note (*frame*, *onset*, *offset*, and *velocity*). It is a transformer-based model consisting of two transformer encoders that work on different axes of the input and a transformer decoder in the middle of these two encoders. Following the processing pipeline in Section 4, we attempted injecting the SONIDO, MUSICGEN, and Jukebox features before the 1st encoder, 2nd encoder, and decoder. We found that feature injection before the decoder yields the best result, thus we adopt this injection method in the following experiments. All the training hyperparameters were kept the same as in a previous study (Toyama et al., 2023). Further details are provided in Appendix D.3.

Following the common evaluation practice (Gardner et al., 2022; Toyama et al., 2023), we report four F1 scores: frame-wise, note-wise, note-wise with offset, and note-wise with offset and velocity using the checkpoint with the best validation F1 score. As shown in Table 4, injecting either SONIDO, MUSICGEN SMALL, or Jukebox features improves the performance of hFT-Transformer. The performance gap is especially huge when the model is trained with a small subset of MAPS. This demonstrates the usefulness of injecting music foundation model features into downstream task models when training data are scarce. We also observed that the decrement of training loss is faster when either SONIDO, MUSICGEN, or Jukebox features are injected, as shown in Figure 8 in Appendix D.3.

In the experiment involving the full MAPS, injecting top and middle SONIDO features yields performance improvement. However, no improvement is observed when all the features from three layers are injected. A similar trend can be observed from the results of MUSICGEN LARGE. We assume that the network capacity required to interpret all the information contained in the features could exceeded that of hFT-Transformer, negatively impacting the model. This suggests that, disentangling feature information on the basis of information granularity to filter out irrelevant information is crucial for such injection.

Table 5: Music source separation. Evaluation results on MUSDB18 and MDXDB21.

| Model | MUSDB18 (BSSEval v4 SDR (dB)) | | | | | MDXDB21 (global SDR (dB)) | | | | |
|---|---|---|---|---|---|---|---|---|---|---|
| | Bass | Drums | Other | Vocals | Average | Bass | Drums | Other | Vocals | Average |
| Open-Unmix (UMX) | 4.01 | 4.35 | 2.79 | 5.66 | 4.20 | 4.50 | **4.46** | 2.66 | 5.55 | 4.29 |
| UMX + MusicGen Small | 4.25 | **4.55** | **3.18** | 5.66 | **4.41** | 4.61 | 4.29 | 2.92 | 5.42 | 4.31 |
| UMX + MusicGen Large | 3.97 | 4.25 | 3.13 | 5.28 | 4.16 | 4.55 | 4.04 | **2.95** | 5.35 | 4.22 |
| UMX + SoniDo | **4.37** | 4.16 | 3.00 | **5.91** | 4.36 | **4.71** | 4.43 | 2.64 | **5.69** | **4.37** |
| HTDemucs (default) | 8.94 | 8.22 | 5.55 | 7.56 | 7.57 | 7.86 | 7.89 | 5.09 | 7.70 | 7.13 |
| HTDemucs (ablation 1) | 8.81 | 8.20 | 5.70 | 7.69 | 7.60 | 7.94 | 7.97 | 5.16 | 7.91 | 7.24 |
| HTDemucs (ablation 2) | 8.75 | 8.64 | 5.78 | 7.85 | 7.76 | 7.96 | 7.69 | 5.12 | 7.89 | 7.17 |
| HTDemucs + STFT-2048 | 5.65 | 6.22 | 4.45 | 6.56 | 5.72 | 5.84 | 6.13 | 4.40 | 6.85 | 5.80 |
| HTDemucs + STFT-4096 | 6.44 | 6.25 | 4.29 | 6.28 | 5.81 | 6.18 | 6.19 | 4.43 | 6.83 | 5.91 |
| HTDemucs + CLAP | 8.25 | 7.37 | 5.21 | 7.21 | 7.01 | 7.37 | 7.51 | 4.82 | 7.47 | 6.79 |
| HTDemucs + MusicGen Small | 8.86 | 8.03 | 5.59 | 7.57 | 7.51 | 7.44 | **8.31** | **5.26** | 7.81 | 7.21 |
| HTDemucs + MusicGen Large | 8.17 | 7.50 | 5.54 | 7.66 | 7.22 | 7.40 | 7.37 | 4.93 | 7.73 | 6.86 |
| HTDemucs + Jukebox | 7.12 | 6.65 | 4.77 | 6.84 | 6.35 | 6.58 | 6.58 | 4.59 | 7.12 | 6.22 |
| HTDemucs + SoniDo | **9.50** | **8.65** | **5.91** | **8.07** | **8.03** | **8.14** | 8.16 | 5.21 | **8.04** | **7.39** |
| HTDemucs (trained on SDXDB23_Bleeding) | 3.86 | 5.52 | 3.53 | 5.70 | 4.65 | 6.20 | 5.98 | 4.53 | 6.69 | 5.85 |
| HTDemucs + SoniDo (trained on SDXDB23_Bleeding) | **5.50** | **6.06** | **3.97** | **5.82** | **5.43** | **6.41** | **6.40** | **4.64** | **7.19** | **6.16** |

### 4.2.2 Music Source Separation: UMX, HTDemucs

We select Open-Unmix (UMX) (Stöter et al., 2019) and Demucs (HTDemucs) (Rouard et al., 2023) for feature injection. UMX estimates the time-frequency mask of the target source using recurrent neural network (RNN) blocks. HTDemucs is a hybrid model with waveform U-Net branch and spectral U-Net branches. We inject the extracted features into the encoder block for UMX using a down-sampling block and in each HTDemucs branch using a cross-domain Transformer (details in Appendices E.1 and E.3). Based on the observation in Section 4.2.1 and for simplicity, we inject only top-level features from SoniDo.

Table 5 lists the SDR scores on the test split of MUSDB18 Rafii et al. (2017) and the hidden split of MDXDB21 Mitsufuji et al. (2022); Fabbro et al. (2023). The details of the experiments are provided in Appendix E. Similar to the experiment discussed in Section 4.2.1, a faster loss decrement is observed, as shown in Figures 12 and 13 in Appendix E. Injecting the SoniDo features into both UMX and HTDemucs greatly boosts the separation performance for both models, even on the unseen dataset MDXDB21. It also improves the separation performance of HTDemucs when training on data corrupted by bleeding errors (SDXDB23_Bleeding) Fabbro et al. (2023). However, the MusicGen features do not always improve the results. Injecting the MusicGen Small features improved UMX, but not for the other cases. According to the ablation study results, injecting short-term Fourier transform (STFT) signal, CLAP features, MusicGen features, or Jukebox features leads to unstable training. However, no such behavior is observed when injecting the SoniDo features into HTDemucs. We assume that performance can be improved if instability during training is avoided. As mentioned in Section 4.2.1, interpreting information contained in MusicGen Large could cost too much capacity of the downstream model and result in performance degradation.

In summary, we observed that injecting the SoniDo features into separation models not only yields faster training and better performance but also improves the robustness to dataset corruption.

### 4.2.3 Music Mixing: Mix-Wave-U-Net, CRAFx2

Mix-Wave-U-Net (Steinmetz et al., 2022) along with a modified CRAFx (Martínez-Ramírez et al., 2020), henceforth referred to as CRAFx2, are used as the baselines. The input to both networks is the stereo stems pre-processed by Fx-normalization (Martínez-Ramírez et al., 2022), and the output is the stereo mixture. These models do not handle high-level information relevant to mixing, such as genre, instrumentation, or mood. Conditioning these models with extracted features, which implicitly contain such information, is expected to improve mixing performance. The features are computed from the monaural downmix of the mixture, which corresponds to the summation of the Fx-normalized input stems. To incorporate these features, we condition both networks using Feature-wise Linear Modulation (FiLM) layers (Perez et al.,

Table 6: Music mixing. Evaluation results on the MDXDB21-dry and MUSDB18 test sets include mean absolute percentage error for audio effect-related features, their average, and stereo-invariant loss. More details are provided in Table 15.

| Model | MDXDB21-dry test set | | | | | | MUSDB18 test set | | | | | |
|---|---|---|---|---|---|---|---|---|---|---|---|---|
| | Spectral | Panning | Dynamic | Loudness | Average | Stereo Invariant | Spectral | Panning | Dynamic | Loudness | Avg | Stereo Invariant |
| Mix-Wave-U-Net (default) | 0.234 | 0.215 | 0.073 | 0.168 | 0.173 | 89.631 | 0.201 | 0.164 | 0.085 | 0.167 | 0.154 | 34.253 |
| Mix-Wave-U-Net + Jukebox | 0.240 | 0.231 | 0.075 | 0.154 | 0.175 | 83.717 | 0.206 | 0.187 | 0.082 | **0.157** | 0.158 | 32.573 |
| Mix-Wave-U-Net + MusicGen Small | 0.240 | 0.197 | **0.064** | 0.147 | 0.162 | 80.161 | 0.214 | **0.158** | 0.079 | 0.163 | 0.153 | 32.151 |
| Mix-Wave-U-Net + MusicGen Large | 0.241 | 0.231 | 0.066 | 0.145 | 0.171 | 81.161 | 0.205 | 0.192 | 0.075 | 0.167 | 0.160 | 32.649 |
| Mix-Wave-U-Net + SONIDO | **0.226** | **0.180** | 0.067 | **0.131** | **0.151** | **78.180** | **0.186** | 0.175 | **0.063** | 0.179 | **0.151** | **30.116** |
| CRAFx2 (default) | **0.193** | 0.179 | 0.070 | 0.152 | **0.148** | 82.095 | 0.193 | **0.154** | 0.081 | **0.165** | 0.148 | 32.856 |
| CRAFx2 + Jukebox | 0.231 | 0.249 | 0.075 | 0.144 | 0.175 | 87.973 | 0.216 | 0.220 | 0.082 | **0.165** | 0.171 | 36.172 |
| CRAFX2 + MusicGen Small | 0.229 | 0.244 | 0.072 | 0.148 | 0.173 | 87.273 | 0.211 | 0.204 | 0.083 | 0.178 | 0.169 | 36.418 |
| CRAFX2 + MusicGen Large | 0.228 | 0.219 | 0.073 | **0.132** | 0.163 | 87.318 | 0.224 | 0.206 | 0.080 | 0.175 | 0.171 | 36.519 |
| CRAFX2 + SONIDO | 0.221 | **0.175** | **0.064** | 0.171 | 0.158 | **79.861** | **0.187** | **0.154** | **0.076** | 0.169 | **0.146** | **30.155** |

2018). For Mix-Wave-U-Net, we inject features into the up-sampling and bottleneck one-dimensional (1D) convolutional blocks. For CRAFx2, we use FiLM layers to condition both the latent-space mixer and synthesis back-end (see Appendix F.1).

We train all models on MUSDB18, and the stereo-invariant loss, along with all training hyperparameters, remains the same, as in a previous study (Martínez-Ramírez et al., 2022). Due to the inherent subjectivity of the task, identifying the best model is challenging. Thus, as shown in Table 1, an objective evaluation is conducted by measuring the proximity between the output mixes and target mixes on the same test sets as in Martínez-Ramírez et al. (2022). The proximity measurement is based on objective metrics (Steinmetz et al., 2022; Martínez-Ramírez et al., 2022). These metrics consist of spectral, panning, dynamic, and loudness low-level audio features, which are the key audio characteristics often manipulated during the mixing process. Further details and experiments are provided in Appendices F.2 and F.3, respectively.

As shown in Table 6, conditioning both architectures with the SONIDO features improved objective performance. The training and validation curves in Figure 14 of Appendix F.3 show a faster loss decrease in early epochs and better generalization, respectively. Although there is no standardized objective evaluation due to the subjective nature of the task (Steinmetz et al., 2022), the presented metrics suggest that the best-performing model closely aligns with the target mixes, resembling professional human-made mixes.

Using the SONIDO features leads to the best performance. The Jukebox and MUSICGEN features provide improvements but not as effective as those of SONIDO, particularly for CRAFx2, where the default model outperforms both Jukebox and MUSICGEN in various metrics. For MUSICGEN, we assume this performance gap may be attributed to its training on 32 kHz audio compared with the 44.1 kHz used for SONIDO, which limit its effectiveness for full-band tasks, such as music mixing, that require higher sampling rates. There is no data-driven approach that used task-agnostic features of the input stems for music mixing improvement. Thus, we can conclude that incorporating the SONIDO features benefits both the training and performance of automatic music mixing models. This aligns with recent design studies (Lefford et al., 2021; Vanka et al., 2023), advocating for the incorporation of contextual inputs.

# 5 Ethical Concerns

To train SONIDO, we acquired an internal dataset of library music with licensing explicitly allowing machine learning training. The dataset is mostly non-vocal, biased toward orchestral and western music. A model trained on this dataset is unlikely to characterize equally well for all types of music. The learnt intermediate embedding may reflect the bias. When using such a biased music foundation model as a performance booster, thorough verification is required before using such a model for practical use or the decision process.

## 6    Conclusions

We extended the use of music foundation models from MIR to generic music downstream tasks. The task-agnostic intermediate representation extracted using proposed SONIDO model has been applied to task-specific models of music tagging, music transcription, music source separation, and music mixing. On the basis of the evaluation results, performance improvement is observed for all selected music downstream tasks. This suggests that incorporating the intermediate features extracted from a pre-trained auto-regressive music foundation model should be considered as a generic booster in future development of task-specific models. This is especially helpful when it is difficult to acquire a sufficient dataset or the computation resource does not allow large-scale training. A study on the bias propagation of a pre-trained music foundation model to a downstream task model should be conducted in another future work.

## Acknowledgement

We would like to thank Takashi Shibuya, Naoya Takahashi, Marc Ferras, and Masato Ishii for many helpful comments during the preparation of this manuscript. Besides, we thank anonymous reviewers for their valuable suggestions and comments.

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

# A    Related Work

## A.1    Understanding models

Understanding models based on supervised learning (SL) has shown good performance on music tagging tasks (Zhong et al., 2023a; Koutini et al., 2022; McCallum et al., 2022), but they have difficulties into addressing tasks that involve unseen annotations during inference (McCallum et al., 2022; Niizumi et al., 2022).

Self-supervised learning (SSL), however, learns features without human annotation. For example, contrastive learning reflects data similarity and dissimilarity in the learned representation (Spijkervet & Burgoyne, 2021; McCallum et al., 2022). Masked reconstruction asks the model to predict the missing data and has been widely applied to language (Devlin et al., 2018), speech (Hsu et al., 2021), sound (Niizumi et al., 2022), and music tasks (Chen et al., 2023). MERT (Li et al., 2023b) further extended masked reconstruction to both acoustic and music features.

SSL methods achieved high performance across a wider range of tasks than SL methods. While a comprehensive set of downstream tasks, including music tagging (Li et al., 2023b; McCallum et al., 2022; Niizumi et al., 2022), music source separation (Chen et al., 2023; Li et al., 2023b) and music bandwidth extension (Zhong et al., 2023b), has been examined, some tasks related to music production, such as music transcription and music mixing, have not been well-investigated.

### A.2 Generative models

Most understanding models are encoder-only models, in which generation is not covered. In contrast, generative models have the ability of generation as well as learning representation (Chen et al., 2020; Dhariwal et al., 2020). Some generative models are based on auto-regressive modeling, thus can execute not only generation but tasks such as music continuation (Dhariwal et al., 2020; Agostinelli et al., 2023; Copet et al., 2023). Recent diffusion-based models can even execute music inpainting (Forsgren & Martiros, 2022; Li et al., 2023a; Huang et al., 2023) and music style transfer (Liu et al., 2023). Multi-tasking can also be achieved by task-augmentation (Li et al., 2023a) or explicit design (Yang et al., 2023). Generation can be conditioned by text, lyrics, and melody (Copet et al., 2023; Agostinelli et al., 2023; Dhariwal et al., 2020). The use of generative models greatly extends the coverage of applicable music downstream tasks.

### A.3 Neural tokenizer in generative models

Vector quantization (VQ) (van den Oord et al., 2017) has been a promising step for music generative modeling. Tokens obtained by VQ can be better characterized with generative models than raw signals. Building music foundation models on top of tokenizers has become a mainstream approach (Agostinelli et al., 2023; Copet et al., 2023).

VQ-VAE-2 (Razavi et al., 2019) first extended VQ learning to hierarchical discrete representation learning in computer vision, which prompted the emergence of the pioneering music generation model (Dhariwal et al., 2020). It was shown to learn global and local information in top and bottom levels, respectively. Another variant of VQ aimed at learning structured discrete representation is residual VQ, which assigns multiple codes to encoded vectors in a residual manner (Zeghidour et al., 2022; Lee et al., 2022). Residual VQ was initially proposed for neural audio codec (Zeghidour et al., 2022; Défossez et al., 2023) and shown to result in coarse-to-fine representation (Lee et al., 2022).

HQ-VAE was proposed to encompass two advanced VQ schemes including RVQ within the variational Bayes framework (Takida et al., 2024). The unified scheme enhances the codebook utilization of the VQ-based models due to the effect of self-annealing (Takida et al., 2022). The authors constructed a hierarchical latent space with three levels and jointly trained the levels on an audio dataset. The model achieved local-to-global representation similar to VQ-VAE-2, thus being freed from layer collapse.

## B Architecture of SONIDO

SONIDO is a generative model that generates music samples conditioned on given text prompts. We train the model using an internal dataset contains around 115k studio-quality library music tracks sampled at 44.1 kHz. Their lengths vary from 30s to 150s, their tempos vary between 50bpm to 200bpm, and the total length is around 4,000h. 90% of the dataset is non-vocal. Although there are more than 50 genres included in the dataset, it is biased toward orchestral and western music. To generate music samples, we use the three priors in SONIDO consecutively from the top level to the bottom level. It is important to note that

during music generation, we use the token sequences sampled from top and middle priors for conditioning. For training and feature extraction, however, we rely on ground-truth tokens.

Given a text prompt, we first obtain the CLAP embedding $\boldsymbol{y}_{text} = \text{CLAP.text\_embedding}(x)$. Conditioned on $\boldsymbol{y}_{text}$, the top prior transformer generates a token sequence $\boldsymbol{Z}_1$ in an auto-regressive manner. Subsequently, $\boldsymbol{Z}_1$ is fed into the middle conditional transformer along with $\boldsymbol{y}_{text}$ to generate $\boldsymbol{Z}_2$. The length of $\boldsymbol{Z}_2$ is four times that of $\boldsymbol{Z}_1$. Similarly, we generate $\boldsymbol{Z}_3$ using the bottom conditional prior by conditioning it with $\boldsymbol{Z}_1$, $\boldsymbol{Z}_2$ and $\boldsymbol{y}_{text}$. The length of $\boldsymbol{Z}_3$ is once again four times that of $\boldsymbol{Z}_2$. Finally, all the three token sequences $\boldsymbol{Z}_{1:3}$ are fed into the decoder of the stage-1 model for audio reconstruction.

## B.1  Stage-1 model

The stage-1 model for SONIDO is based on the aforementioned SQ-VAE-2 structure, as illustrated in Figure 2(a). It is a three-level SQ-VAE-2 (i.e., $L = 3$) autoencoder. It comprises three encoding blocks, denoted as $encoder_{bottom}$, $encoder_{middle}$, and $encoder_{top}$, where the *bottom* layer processes the input audio sampled at 44.1 kHz. Each encoding block consists of 1-D convolutional layers with a strided convolution for down-sampling. The down-sampling ratios are set to 8, 4, and 4 for $encoder_{bottom}$, $encoder_{middle}$, and $encoder_{top}$, respectively. The stage-1 model has three decoding blocks, $decoder_{bottom}$, $decoder_{middle}$, and $decoder_{top}$. Each is a mirrored version of the encoding block with the same resolution. We train the stage-1 model on top of the HQ-VAE framework on $4,000$h of 44.1-kHz studio-quality library music. The token sequence generated by the stage-1 model is used to train the prior $P(\boldsymbol{Z}_{1:3})$, as presented in the subsequent section.

## B.2  Stage-2 model

For the stage-2 model, we followed Jukebox (Dhariwal et al., 2020), which generates hierarchical tokens with the same down-sampling rates as the stage-1 model (i.e, a token in the top/middle/bottom level compresses $128/32/8$ audio samples). Specifically, we train three transformers in an auto-regressive manner to model $P(\boldsymbol{Z}_1|\boldsymbol{y}_{\text{audio}})$, $P(\boldsymbol{Z}_2|\boldsymbol{Z}_1, \boldsymbol{y}_{\text{audio}})$, and $P(\boldsymbol{Z}_3|\boldsymbol{Z}_{1:2}, \boldsymbol{y}_{\text{audio}})$, which we call the top prior, middle conditional prior, and bottom conditional prior, respectively.

Initially, we train a sparse transformer to learn the probability distribution of top-level token sequence $\boldsymbol{Z}_1$ in an auto-regressive manner, conditioned on $\boldsymbol{y}_{\text{audio}}$ (i.e., $P_{\boldsymbol{\pi}_1}(\boldsymbol{Z}_1|\boldsymbol{y}_{\text{audio}})$). Subsequently, we train other sparse transformers to model $P_{\boldsymbol{\pi}_i}(\boldsymbol{Z}_l|\boldsymbol{Z}_{<l}, \boldsymbol{y}_{\text{audio}})$. These transformers are conditioned by token sequences from upper levels. We used similar configurations of Jukebox's transformers, but modified some hyperparameters, as shown in Table 7. The primary distinction between them lies in the conditioning mechanism for each prior. We modified conditioning modules to use $\boldsymbol{y}_{\text{audio}}$, instead of the original conditioning inputs employed in Jukebox such as artists, genres, and lyrics. In contrast to Jukebox, the bottom conditional transformer in our model is conditioned on all the upper token sequences $\boldsymbol{Z}_{<l}$, diverging from Jukebox's approach, where the $l^{th}$ transformer is conditioned on the adjacent upper token sequences $\boldsymbol{Z}_{l-1}$. This difference is a result of the hierarchical structure obtained with SQ-VAE-2, which has a tight interrelation between different levels, unlike Jukebox's independently trained multi-level token sequences.

We explain the entire conditioning mechanism using the bottom conditional prior as an example, which allows us to address all the detail (see Figure 2). We first obtain the hierarchical discrete representations $\boldsymbol{Z}_1$, $\boldsymbol{Z}_2$, and $\boldsymbol{Z}_3$ by applying the pre-trained stage-1 model to an input musical audio $x$. The goal of training is to make the model execute next-token prediction on $\boldsymbol{Z}_3$, conditioned on $\boldsymbol{Z}_1$, $\boldsymbol{Z}_2$ and the CLAP embedding $\boldsymbol{y}_{audio} = CLAP.audio\_embedding(x)$. We condition the transformer in a frame-by-frame manner. Since each level has a different time resolution, each token sequence from upper level is first converted into embedding sequence and then up-sampled to the next time resolution by a transposed convolution. We use two up-sampling modules: one for $\boldsymbol{Z}_1 \rightarrow \boldsymbol{Z}_2$ and the other for $\boldsymbol{Z}_1, \boldsymbol{Z}_2 \rightarrow \boldsymbol{Z}_3$. The conditioning module $\boldsymbol{Z}_1 \rightarrow \boldsymbol{Z}_2$ up-samples the embedding sequence $\boldsymbol{Z}_1$ to match the resolution of $\boldsymbol{Z}_2$. Similarly, up-sampling module $\boldsymbol{Z}_1, \boldsymbol{Z}_2 \rightarrow \boldsymbol{Z}_3$ takes $\boldsymbol{Z}_2$ as input, along with the up-sampled embeddings of $\boldsymbol{Z}_1$, generating further up-sampled embeddings tailored to $\boldsymbol{Z}_3$'s resolution. The resulting frame-level embedding is used to condition the transformer for $\boldsymbol{Z}_3$.

For the actual next-token prediction task, we shift the token sequence $\boldsymbol{Z}_3$ by one position to the right and embed each token to a continuous vector using an embedding layer. The first token, which is empty, is

Table 7: Hyper-parameter comparison on Jukebox's and SONIDO's top-level prior. Hyper-parameter marked with * indicates that we used different setting from Jukebox. Otherwise we used same configuration.

| | Jukebox's | | | SONIDO's | | |
| --- | --- | --- | --- | --- | --- | --- |
| | Top | Middle | Bottom | Top | Middle | Bottom |
| Sample length | 1048576 | 262144 | 65536 | 1048576 | 262144 | 65536 |
| Context length | 8192 | 8192 | 8192 | 8192 | 8192 | 8192 |
| Transformer width | 4800 | 1920 | 1920 | 4800 | 3200 | 2880 |
| Transformer self-attention layers | 72 | 72 | 72 | 72 | 72 | 72 |
| Attention heads | 8 | 1 | 1 | 8 | 4 | 4 |
| Factorized attention shape | (128, 64) | (128, 64) | (128, 64) | (128, 64) | (128, 64) | (128, 64) |
| Encoder-decoder attention layers | 7 | - | - | 7 | - | - |
| Up-sampling modules | - | 1 | 1 | - | 1 | 2 |
| Up-sampling-module residual block width | - | 1024 | 1024 | - | 1024 | 1024 |
| Up-sampling modules residual blocks | - | 16 | 16 | - | 16 | 16 |
| Up-sampling-module conv filter size | - | 3 | 3 | - | 3 | 3 |
| Up-sampling-module conv channels | - | 1024 | 1024 | - | 1024 | 1024 |
| Up-sampling-module dilation growth rate | - | 3 | 3 | - | 3 | 3 |
| Up-sampling-module dilation cycle | - | 8 | 8 | - | 8 | 8 |
| Initialization scale | 0.002 | 0.004 | 0.008 | 0.002 | 0.004 | 0.004 |
| Encoder initialization scale | 0.014 | - | - | 0.014 | - | - |
| Batch size* | 512 | 192 | 184 | 240 | 240 | 240 |
| Training step* | 310500 | 265000 | 279000 | 88000 | 152000 | 272000 |
| Learning rate* | 0.00015 | 0.0003 | 0.0003 | 0.0001 | 0.0002 | 0.0002 |
| Adam $\beta_2$ | 0.925 | 0.95 | 0.95 | 0.925 | 0.95 | 0.95 |
| Weight decay | 0.002 | 0.01 | 0.01 | 0.002 | 0.01 | 0.01 |

optionally filled with $\boldsymbol{y}_{audio}$ to make the system conditioned on the CLAP embedding. With the frame-level aggregated conditioning vectors, namely, up-sampled embeddings from the upper layers, $\boldsymbol{y}_{audio}$, and the time positional embeddings, the transformer is trained to estimate $\boldsymbol{Z}_3$ in an auto-regressive manner.

## B.3 Evaluation on Music Generation

For evaluation, we used the MusicCaps dataset (Agostinelli et al., 2023), which includes 5,521 [2] pairs of audio clips and their text captions written by musicians. Subsequently, we used SONIDO to generate 10-s audio samples, conditioned upon textual captions of MusicCaps. Following the objective evaluation methodology(Copet et al., 2023), we evaluated SONIDO's performance on MusicCaps using three objective metrics: the Fréchet audio distance (FAD) (Kilgour et al., 2019), Kullback-Leibler Divergence (KL), and CLAP (Wu* et al., 2023) score.

Table 8 compares SONIDO in terms of the three metrics with other SOTA auto-regressive methods, namely MusicLM (Agostinelli et al., 2023), MeLoDy (Lam et al., 2023), and MusicGen (Copet et al., 2023).

We used FAD to measure the overall quality of the generated samples. It is a reference-free metric that compares embedding statistics computed on an evaluation set with embedding statistics computed on the clean dataset. A lower FAD indicates a higher degree of similarity between the embedding statistics from generated audio samples and the embedding statistics from the ground-truth (i.e., MusicCaps). We used the official VGGish based-FAD computation module[3] to extract embeddings from audio. Since VGGish was trained using 16-kHz audio, all the audio data were resampled to 16 kHz.

We also computed KL-divergence, which compares the two probability distributions of labels. The labels are estimated by applying a pre-trained audio classifier to the generated and original audio clips. Following (Copet et al., 2023), we used a SOTA audio classifier called PaSST. A lower KL-divergence indicates that audio content generated from a model aligns more closely with the characteristics of the reference. We report the mean of KL-divergence.

---

[2]We used a subset of 5,514 samples due to the presence of expired or invalid URLs in MusicCaps items. Efforts were made to retrieve the missing files; however, three samples are absent despite our efforts.

[3]`https://github.com/google-research/google-research/tree/master/frechet_audio_distance`

Table 8: Objective evaluation of SONIDO and other auto-regressive generative models on MusicCaps. Model marked with * indicates that we generated audio samples and computed metrics using same protocol.

| | MUSICCAPS Test Set | | | |
| Model | Target SR | $\text{FAD}_{vgg} \downarrow$ | $\text{KL} \downarrow$ | $\text{CLAP}_{scr} \uparrow$ |
|---|---|---|---|---|
| MusicLM | 24kHz | 4.0 | - | - |
| MeLoDy | 24kHz | 5.41 | - | - |
| MusicGen-large | 32kHz | 3.8 | 1.22 | 0.31 |
| MusicGen-large (public [4])* | 32kHz | 5.88 | 1.41 | 0.27 |
| SONIDO * | 44.1kHz | 4.98 | 1.45 | 0.39 |

Finally, we calculated the CLAP consistency loss to provide a consistent benchmark aligned with SOTA methodologies, even though it might not be entirely fair since SONIDO was trained using CLAP embeddings. We computed the cosine similarity between the audio CLAP embedding from the generated clip and text CLAP embedding from the original text caption from MusicCaps. A higher CLAP score indicates a stronger alignment between the generated clip and original text description.

SONIDO achieved an FAD (denoted as $\text{FAD}_{vgg}$) of 4.98, a performance comparable to that of the other models. We only report the KL-divergence computed using PaSST as the audio classifier in Table 8. SONIDO attained a KL Divergence (denoted as KL) of 1.45, which is slightly higher than MusicGen-large. Finally, we report the CLAP consistency score (denoted as $\text{CLAP}_{scr}$). SONIDO's CLAP score was the highest, but it should be noted that SONIDO was trained using CLAP embeddings, as we mentioned earlier.

While the evaluation result of SONIDO is slightly inferior to the SOTA, it is important to note that SONIDO generates audio with a higher sampling rate. Table 8 shows the target sampling rate of each model denoted as TARGET SR. While SONIDO generated 44.1 kHz directly, the other models generated audio with lower sampling rates. Furthermore, the evaluation protocol requires SONIDO to be evaluated in a much lower sampling rate. Despite this challenge, we designed SONIDO to operate at a studio quality level (i.e., 44.1 kHz), enabling its potential use in most downstream tasks, which we describe in the following sections.

## C    Music Tagging

We implemented the attention-based aggregator described in Section 3.2 as a 1-layer standard transformer. There is only single attention head in the attention layer. Features from different priors are processed by an input normalization layer then converted to 768 dimensions with a linear layer.Similar to Castellon et al. (2021), we conducted a grid search for the method of input normalization (BatchNorm or LayerNorm), and the dropout ratio in MLP (0.10, 0.25, 0.50, 0.75). For all single-label tasks, by default we set the label smoothing to 0.1. The batch size was set to 256 for MTAT and Nsynth for their large amounts of data, and 64 for others. Unless specifically mentioned, the learning rate was $5\text{e}^{-5}$.

---

[4]Publicly released MusicGen-large trained on non-vocal dataset: `https://github.com/facebookresearch/audiocraft/blob/main/model_cards/MUSICGEN_MODEL_CARD.md`,

Table 9: Datasets for music tagging. Except MTAT (multi-label) and EmoMusic (2-axis regression), all other tasks are single-label. Segment length of VocalSet is after pre-processing.

| Dataset | Task | Num. of Classes | Num. of Segments | Segment Length | Sampling Rate |
|---|---|---|---|---|---|
| MTAT | Auto tagging | 50 | $\sim 25$ k | 29 s | 16 kHz |
| Nsynth | Pitch | 128 | $\sim 306$ k | 4 s | 16 kHz |
| | Instrument | 11 | | | |
| EmoMusic | Emotion regression | n/a | $\sim 740$ | 45 s | 44.1 kHz |
| GiantSteps | Key estimation | 24 | $\sim 2.1$ k | 120 s | 44.1 kHz |
| GTZAN | Genre | 10 | $\sim 900$ | 30 s | 22.05 kHz |
| VocalSet | Singer | 20 | $\sim 7.5$ k | 3 s | 44.1 kHz |
| | Vocal technique | 10 | | | |

Table 10: Preliminary study on impact of CLAP conditioning in music tagging. Acc.: accuracy. mAP: mean average precision (**bold**: top score).

| Features | MTAT mAP | Nsynth-pitch Acc. |
|---|---|---|
| CLAP | 39.6 | 48.8 |
| Unconditional SONIDO | 39.0 | 90.6 |
| CLAP-conditional SONIDO | **39.9** | **91.5** |

Table 11: Ablation study on feature aggregation pipeline in music tagging (**bold**: top score)

| Features | Aggregation | MTAT ROC-AUC | mAP |
|---|---|---|---|
| Top | Average pooling | 91.1 | 39.9 |
| Top | Attention | 91.1 | 40.4 |
| Mid. | Average pooling | 91.1 | 39.4 |
| Mid. | Attention | 91.3 | 40.9 |
| Top + mid. | Average pooling | 91.5 | 40.6 |
| Top + mid. | Attention | **91.7** | **41.5** |

### C.1 Comparison of Features

Since SONIDO is trained with an upstream auto-regressive modeling task, where both CLAP-conditional and unconditional cases are introduced, it is still unclear which setting should be used for downstream tasks. We thus conducted a preliminary study with MTAT, a dataset designed for coarse-grained tagging tasks, and Nsynth-pitch, a dataset designed for fine-grained classification tasks. In this experiment, features from the top prior were aggregated by average pooling, and the learning rate was set to $1e^{-4}$. As mentioned above, the CLAP audio encoder receives the same clip of music as SONIDO and encodes it into a single feature vector. The results in table 10 indicates that CLAP performs well for coarse concepts, while unconditional feature extraction results in better accuracy for pitch estimation. On top of them, the CLAP-conditional feature extraction achieved better scores in both tasks. We will thus use CLAP-conditional feature extraction as the default feature extraction for time-invariant retrieval tasks.

### C.2 Investigation on Feature Aggregation

Next, we conducted ablation studies to verify the effectiveness of the feature aggregation pipeline. The learning rate for attention-based and average pooling is $5e^{-5}$ and $1e^{-4}$, respectively. As shown in Table 11, we have the following observations: (1) the feature aggregation pipeline outperformed the average pooing baselines; (2) applying the same pipeline to non-hierarchical features did not bring as much performance gain as the hierarchical case, showing the hierarchical features contain complementary information for music tagging. The bottom prior did not contribute to a better performance. We assume this is because the bottom prior processes high-frequency components which are less important for the tasks we tested. It is also worth mentioning that many SOTA classification models have been working with $F_s = 16$ kHz, *e.g.*, Niizumi et al. (2022); McCallum et al. (2022); Wang et al. (2022), neglecting high frequency components.

## D Music Transcription

Music transcription requires accurate predictions of both pitch and temporal information for optimal performance. Current models (Toyama et al., 2023; Gardner et al., 2022) rely exclusively on spectrograms as input. There have been attempts in applying extra feature to music transcription. For example, Donahue et al. (2022) used Jukebox for melody transcription. However, the HookTheory dataset used by Donahue et al. (2022) does not include octave information, which makes it differ from the typical music transcription. There are two ways of modeling the typical music transcription task: piano-roll-based music transcription (Hawthorne et al., 2018; Kim & Bello, 2019; Gardner et al., 2022; Cheuk et al., 2021b; Thickstun et al., 2016; Kelz et al., 2019; Toyama et al., 2023) and token-based music transcription (Gardner et al., 2022). The former transforms spectrograms into piano rolls in which both have the same time resolution. The

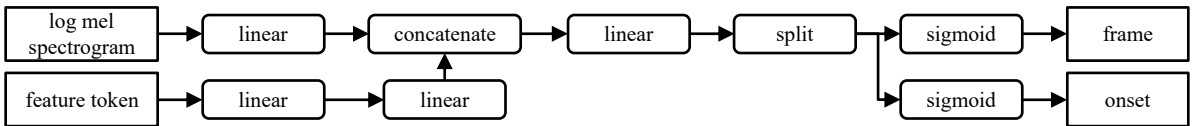

Figure 5: Model architectures of linear music transcription for piano

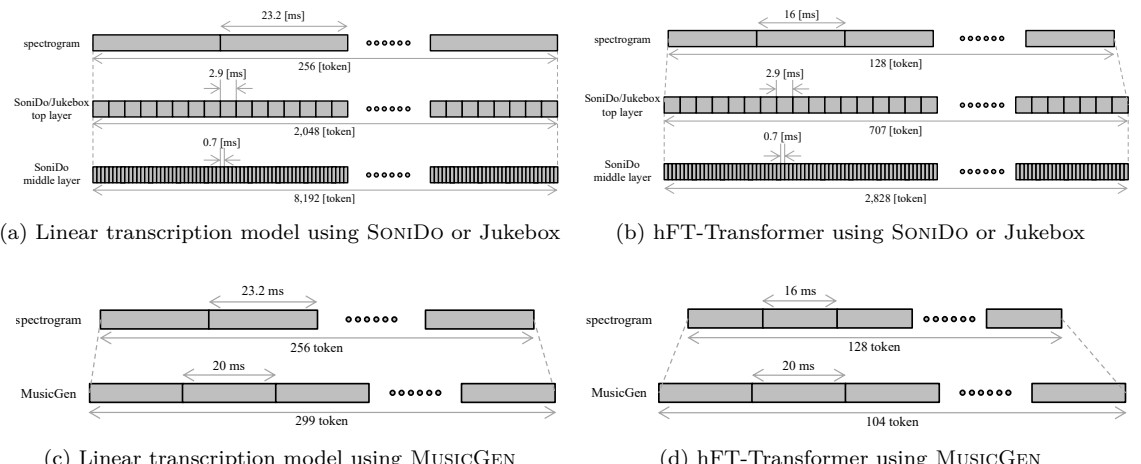

(a) Linear transcription model using SONIDO or Jukebox      (b) hFT-Transformer using SONIDO or Jukebox

(c) Linear transcription model using MUSICGEN      (d) hFT-Transformer using MUSICGEN

Figure 6: Feature alignment schemes for different models in music transcription

latter transforms spectrograms into a series of tokens indicating the note pitch and note on and off locations. We selected the relatively well-established piano-roll-based music transcription to test the effectiveness of intermediate representation extracted from a music foundation model.

## D.1 Piano Transcription with Linear Layers

To understand the usefulness of the extracted intermediate features, we started with piano transcription by probing those features with linear layers. The architecture used for this experiment is shown in Figure 5. The inputs are log mel spectrogram and the SONIDO features. Input audio signals are down-sampled to 22.05 kHz then converted to a 256-bin mel spectrogram using a 2048-point Hann window with a 512 hop size. The alignment between the mel spectrogram and extracted features is illustrated in Figures 6(a) and (c), respectively.

The mel spectrogram is a 3D tensor $(B, N, F)$, where $B = 8$ is the batch size, $N = 256$ is the number of frames, and $F = 256$ is the number of frequency bins. A linear layer converts the tensor into $(B, N, Z)$ along the last axis, where $Z = 512$ is a common size for latent embeddings. The SONIDO, MUSICGEN , and Jukebox features are formed as a 3D tensor $(B, V, G)$, where $V$ is the number of tokens per $N$ frames (2,048 for the top layer of SONIDO and for Jukebox, 8,192 for the middle layer of SONIDO, and 299 for MUSICGEN), and $G$ is the embedding size of features (4,800 for the top layer of SONIDO and for Jukebox, 3,200 for the middle layer of SONIDO, 2,048 for MUSICGEN LARGE, and 1,024 for MUSICGEN SMALL). The first linear layer converts $G$ to $Z$, then the second linear layer converts $V$ to $N$ to make the tensor into the shape of $(B, N, Z)$. The tensors are then concatenated along the last axis. If the top layer of SONIDO, middle layer of SONIDO, or MUSICGEN is used, the dimension of the last axis is $2Z$, whereas if both layers of SONIDO are used, the dimension is $3Z$. The following linear layer is used to convert $2Z$ or $3Z$ to $2P$, where $P = 88$ is the number of pitches. Finally, the tensor is split into two $(B, N, P)$ tensors followed by a sigmoid activation to indicate the estimated *frame* and *onset* information. We followed Kong et al. (2021) to extract the precise timing of onsets from datasets, in which we set the hyper-parameter to control the target sharpness $J$ to 3. The loss function for *frame* and *onset* is binary cross entropy.

Table 12: F1 scores on different datasets when best validation checkpoint was obtained using Bach10 as validation set.

| Dataset | Best validation checkpoint | | | | | |
|---|---|---|---|---|---|---|
| | Base | MusicGen Small | MusicGen Large | SoniDo Top | SoniDo Middle | SoniDo Top+Middle |
| Bach10 | 43.3 | 68.4 | 60.7 | 69.4 | **74.4** | 72.5 |
| GuitarSet | 34.8 | 49.7 | 38.6 | 43.0 | **50.1** | 44.9 |
| Su | 13.5 | **32.7** | 31.7 | 21.1 | 27.3 | 24.4 |
| TRIOS | 20.8 | 38.2 | 31.4 | 34.9 | **40.7** | 39.4 |

We trained multiple models with the following input: (1) spectrogram only, (2) spectrogram and the top layer of the SoniDo features, (3) spectrogram and the middle layer of the SoniDo features, (4) spectrogram and both the top and middle layers of the SoniDo features, (5) spectrogram and the MusicGen Large features, (6) spectrogram and the MusicGen Small features, and (7) spectrogram and the Jukebox features. For each model, we trained for 50 epochs on one A100 graphics processing unit (GPU), using Adam (Kingma & Ba, 2015) optimizer with a learning rate of $1e^{-4}$. `PyTorch ReduceLROnPlateu` was used for learning-rate scheduling with default parameters. We chose to use the checkpoint with the highest F1 score in the validation split for each model.

As listed in Table 3, the model using the SoniDo features, the MusicGen features, or the Jukebox features outperformed the spectrogram-only baseline. In particular, the contribution of SoniDo's middle layer was higher than that of the top layer. These results suggest that the SoniDo features are promising for music transcription tasks.

## D.2 Linear Instrument-agnostic Music Transcription

Following the setup described previously in D.1, we also investigated the effectiveness of the extracted features with instrument-agnostic music transcription. We wanted to study if the features of music foundation model are also applicable to a multi-instrument transcription scenario. In this experiment, we reused the model illustrated in Figure 5 and trained it on the URMP dataset, which contains strings, woodwinds and brass instruments (Li et al., 2019). Unlike piano transcription, we do not need to split the final output into frame and onset prediction for instrument-agnostic transcription. This is due to the fact that some musical instruments, such as violins and flutes, sometimes produce ambiguous onset attacks. Applying onset prediction to these instruments can be detrimental to the F1 score (Cheuk et al., 2021a).

We validated the model performance on the Bach10 dataset (Duan et al., 2010) and selected the best checkpoint based on the validation performance. We then evaluated the best checkpoint on GuitarSet (Xi et al., 2018), Su (Su & Yang, 2016), and TRIOS (Fritsch, 2012) datasets. Table 12 shows the F1 scores obtained using the best checkpoint. The results indicate that features of music foundation model, regardless of SoniDo or MusicGen, boost instrument-agnostic transcription performance compared with the baseline model, which uses only the spectrogram as the input features. This indicates that the model trained using these features has a stronger generalizability across different datasets. SoniDo generally outperformed MusicGen on Bach10, GuitarSet, and TRIOS, while MusicGen performed better on Su. This difference may be due to the different datasets on which SoniDo and MusicGen were trained. We will investigate this further in the future.

## D.3 hFT-Transformer

The detailed model architecture of hFT-Transformer has been described in (Toyama et al., 2023). Figures 6(b) and (d) show the feature-spectrogram alignment scheme for hFT-Transformer. The feature tokens and spectrogram cannot be perfectly aligned due to the different sampling frequencies in hFT-Transformer (16 kHz), SoniDo (44.1 kHz), and MusicGen (32 kHz). The $N$ frame in the spectrogram corresponds to roughly 707 tokens of the top layer of the SoniDo features and of the Jukebox features, $2,828$ tokens of the middle layer of the SoniDo features, and 104 tokens of the MusicGen features. Figure 7 shows the modified hFT-Transformer that accepts the SoniDo or MusicGen features as the additional input. Following the

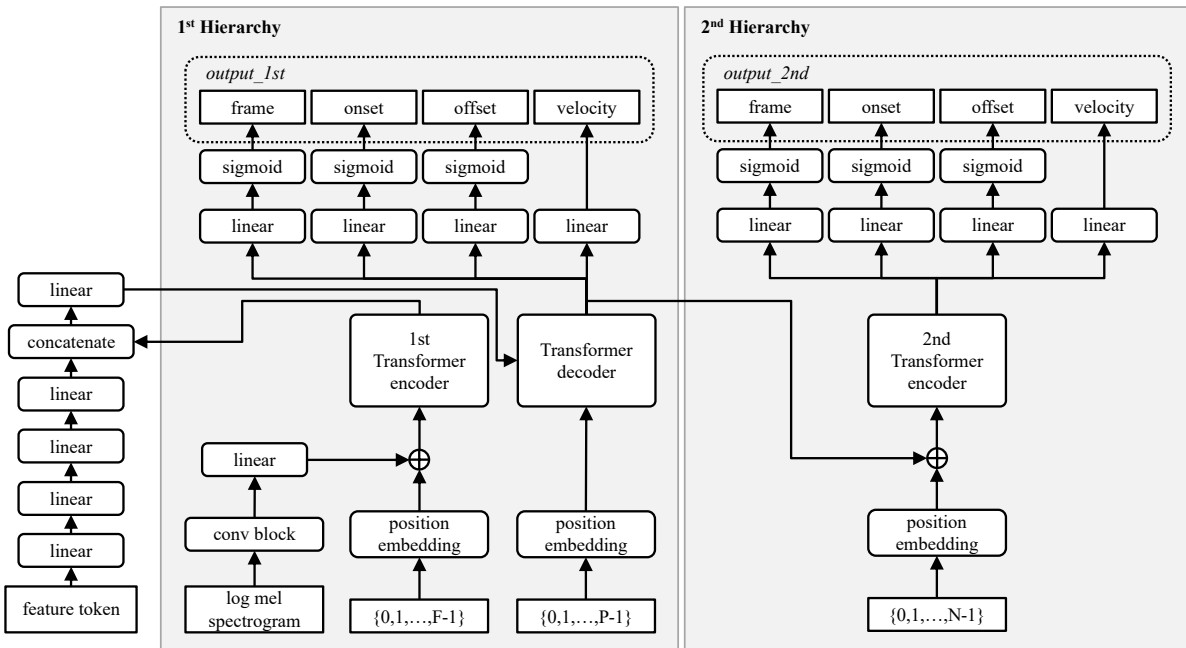

Figure 7: Modified version of hFT-Transformer for SoniDo, MusicGen , and Jukebox feature tokens

setup in D.1, the SoniDo and MusicGen feature tokens have three dimensions $(B, V, G)$, where $B = 8$ is the batch size, $V$ is the number of tokens per number of frames $N = 128$, and $G$ is the embedding size of the feature tokens. The first linear layer for the feature tokens reduces $G$ to $Z' = 256$; the second linear layer reduces $V$ to $N$ then reshapes the tensor to $(B, N, F'' = 16, Z'' = 16)$; the third linear layer changes $Z''$ to $Z = 256$, then the last linear layer changes $F''$ to $F' = 128$. Thus, a tensor with shape $(B, N, F', Z)$ is obtained. When using both the top and middle layers of the SoniDo features, we form such tensors for each layer following the pipeline above. The tensor(s) and output of the first encoder are then concatenated on the third axis. Finally, the size of the concatenated tensor is reduced to 256 ($F$ in (Toyama et al., 2023)) by a linear layer. These $F'$, $F''$, $Z'$, and $Z''$ were determined from preliminary experiments.

We trained the following models that have different inputs: (1) spectrogram only, (2) spectrogram and the top layer of the SoniDo features, (3) spectrogram and the middle layer of the SoniDo features, (4) spectrogram and both the top and middle layer of tthe SoniDo features, (5) spectrogram and the MusicGen Large features, (6) spectrogram and the MusicGen Small features, and (7) spectrogram and the Jukebox features, the same as the experiment described in D.1. We trained the models for 50 epochs on one A100 GPU. For the other conditions, we followed (Toyama et al., 2023). To confirm if these features are useful when there are less training data, we train the models using 100, 50, 25, and 10% of training data. We chose the checkpoint with the highest F1 score in the validation split for further evaluation.

Figure 8 shows the loss curves of training and validation. The models using the SoniDo, MusicGen or Jukebox features reached a lower loss value at an earlier epoch compared with the baseline model using the spectrogram only as input. Table 13 lists the scores on the test set of MAPS. When all the training data were available, the models using the SoniDo, MusicGen , and Jukebox features, except the model using the MusicGen Large features outperformed the baseline. When the models were trained with scarce data, the performance of the models using the SoniDo, MusicGen , and Jukebox features was superior to that of the model using the spectrogram only. When the training data size was 50 and 25%, the performance of the models using the SoniDo features was still comparable to the baseline model trained with 100% data.

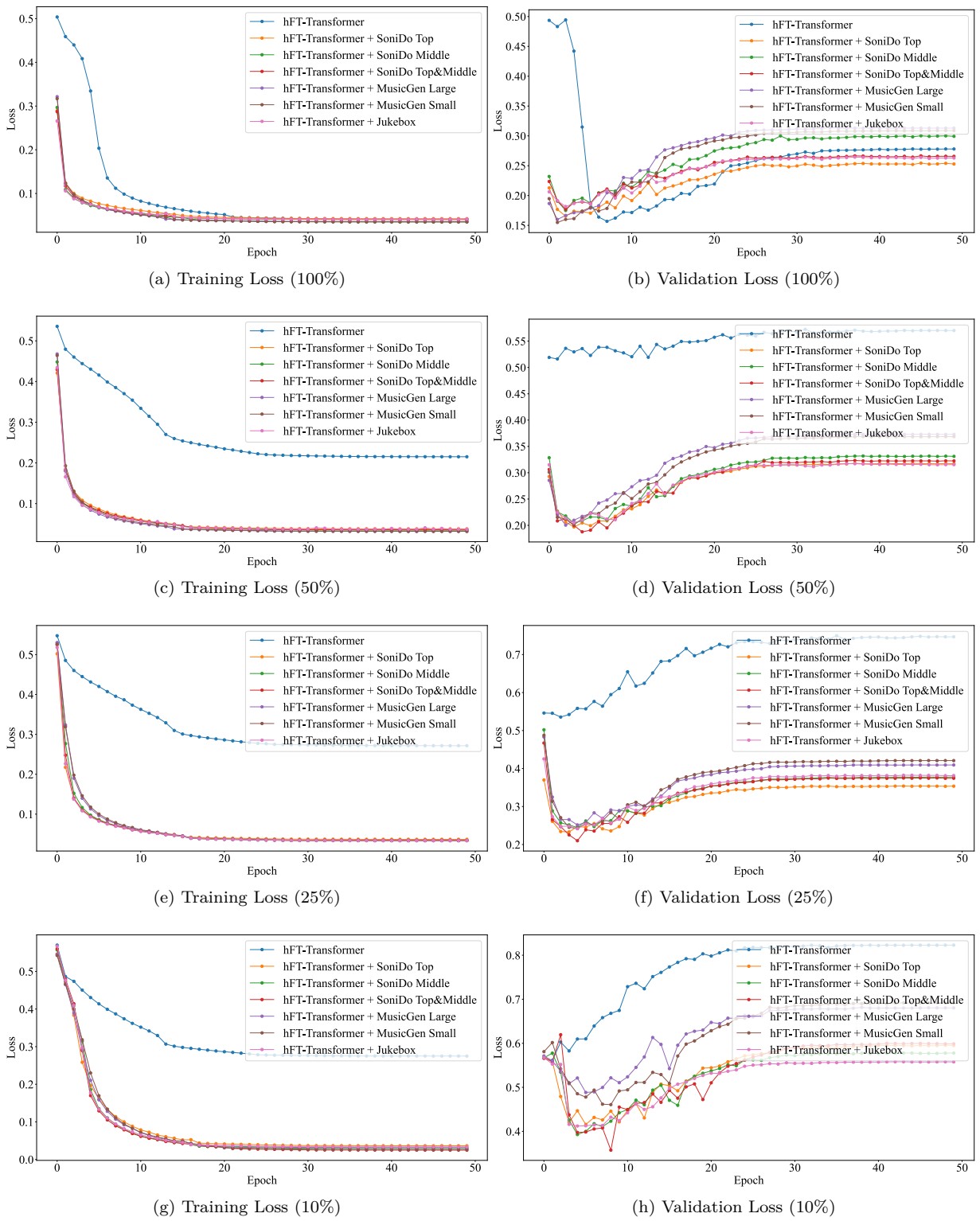

Figure 8: Loss curves for hFT-Transformer in music transcription task

Table 13: Evaluation results on MAPS in music transcription (**bold**: best score, underline: second-best score). "Note" refers to note-wise estimation. First row is of hFT-Transformer Toyama et al. (2023).

| Training data | Input | Frame | Note | Note w/ Offset | Note w/ Offset&Velocity |
|---|---|---|---|---|---|
| | Spectrogram | 82.89 | 85.14 | 66.34 | 48.20 |
| | Spectrogram + SONIDO Top | 83.92 | 86.45 | 68.27 | **51.34** |
| | Spectrogram + SONIDO Middle | 83.47 | 86.13 | 67.93 | 51.24 |
| 100% | Spectrogram + SONIDO Top + SONIDO Middle | 84.16 | 85.96 | 67.37 | 50.98 |
| | Spectrogram + MUSICGEN LARGE | 81.53 | 85.14 | 66.28 | 48.69 |
| | Spectrogram + MUSICGEN SMALL | 82.94 | 85.97 | **68.27** | 50.42 |
| | Spectrogram + Jukebox | **84.23** | **86.54** | 68.26 | 50.46 |
| | Spectrogram | 39.12 | 23.34 | 13.22 | 9.12 |
| | Spectrogram + SONIDO Top | 83.35 | 85.51 | 65.84 | 47.40 |
| | Spectrogram + SONIDO Middle | 82.52 | 85.46 | 65.40 | 47.25 |
| 50% | Spectrogram + SONIDO Top + SONIDO Middle | **83.37** | 85.33 | **67.04** | **49.32** |
| | Spectrogram + MUSICGEN LARGE | 82.26 | 84.58 | 65.50 | 47.07 |
| | Spectrogram + MUSICGEN SMALL | 82.24 | 85.21 | 65.32 | 46.72 |
| | Spectrogram + Jukebox | 83.33 | **85.58** | 64.62 | 46.20 |
| | Spectrogram | 12.88 | 1.61 | 0.66 | 1.01 |
| | Spectrogram + SONIDO Top | 81.71 | **84.70** | 63.00 | **45.50** |
| | Spectrogram + SONIDO Middle | 81.65 | 84.59 | 62.19 | 43.91 |
| 25% | Spectrogram + SONIDO Top + SONIDO Middle | 81.44 | 83.79 | 62.61 | 44.71 |
| | Spectrogram + MUSICGEN LARGE | 78.98 | 82.36 | 58.64 | 39.62 |
| | Spectrogram + MUSICGEN SMALL | 79.39 | 82.23 | 60.36 | 41.02 |
| | Spectrogram + Jukebox | **81.96** | 84.47 | **63.63** | 44.63 |
| | Spectrogram | 9.83 | 0.59 | 0.17 | 0.46 |
| | Spectrogram + SONIDO Top | 65.91 | 66.64 | 39.88 | 25.87 |
| | Spectrogram + SONIDO Middle | 70.77 | 74.02 | 45.75 | 29.46 |
| 10% | Spectrogram + SONIDO Top + SONIDO Middle | **71.57** | **75.00** | **46.18** | **30.63** |
| | Spectrogram + MUSICGEN LARGE | 61.81 | 63.27 | 37.03 | 24.01 |
| | Spectrogram + MUSICGEN SMALL | 63.73 | 65.90 | 39.00 | 24.94 |
| | Spectrogram + Jukebox | 70.43 | 73.76 | 45.80 | 30.42 |

# E  Music source separation

## E.1  Details of UMX with SONIDO

Huang et al. (2022b) investigated various speech enhancement (SE) systems that use self-supervised learning (SSL) features and discussed the challenge that the SSL features may have lost some local signal information necessary for estimating lower-level features (e.g., spectrograms, waveform). Following the above observation, Hung et al. (2022) proposed to combine spectrograms with SSL features in their SE system to avoid such problem. Therefore, we hypothesize that the features extracted with a large-scale foundation model could serve as auxiliary information for a neural network on music source separation. However, it remains unclear how to integrate the extracted features into the network. Therefore, we investigated several integration strategies.

Figures 9 and 10 show the architectures of the original UMX and UMX with SONIDO, respectively. UMX starts with an input audio waveform and converts it into an STFT spectrogram. The magnitude part of the spectrogram is normalized to a mean of 0 and standard deviation of 1, based on statistics collected from the entire training dataset before training. The normalized magnitude spectrogram is then passed through an "encoder" block, which includes a linear layer, batch normalization (BN) layer, and hyperbolic tangent

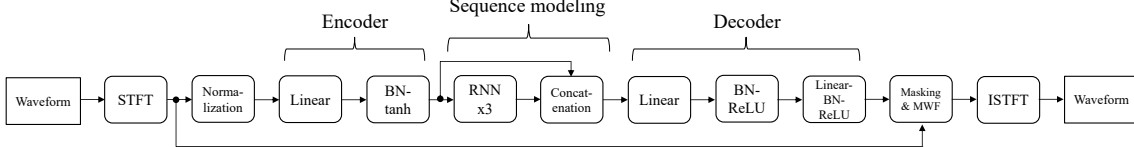

Figure 9: Original architecture of UMX

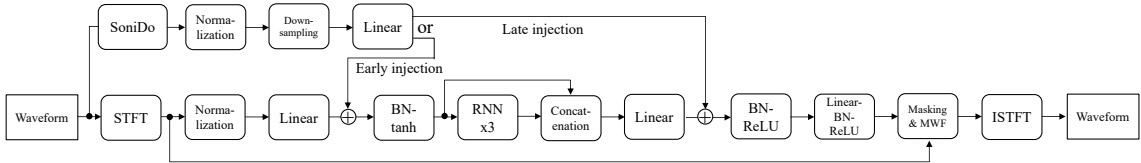

Figure 10: Architecture with SONIDO using UMX

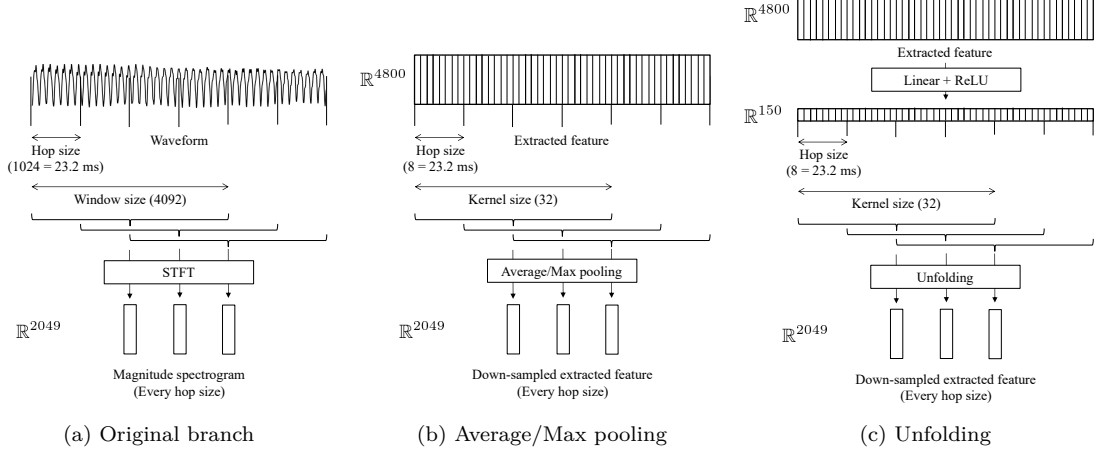

(a) Original branch     (b) Average/Max pooling     (c) Unfolding

Figure 11: Down-sampling blocks to align time resolution.

function. A "sequence modeling" block, consisting of three RNN layers along with a skip connection structure is then used. It processes the encoded features and combines their input and output features. The resulting combined features are then transformed into time-frequency masks in the "decoder" block, which is made up of several linear layers, BN layers, and rectified linear unit (ReLU) activations. These time-frequency masks are used to modify the magnitude spectrogram, and a multi-channel Wiener filter (MWF) Nugraha et al. (2016); Uhlich et al. (2017) is optionally applied to the spectrogram. Finally, an inverse STFT yields the separated audio waveform.

All learnable parameters are optimized with the mean-squared error of magnitude spectrograms. We propose using the SONIDO features while keeping the UMX architecture unchanged. As shown in Figure 10, SONIDO is introduced to extract features from the input audio waveform. These features are then adjusted to have a standard statistical distribution computed across the entire training dataset similarly to the normalization used for the magnitude spectrograms in UMX. Subsequently, they are processed through a down-sampling block, followed by a linear layer. The down-sampling block is introduced because of the time resolution difference between SONIDO and UMX; SONIDO generates a feature for every 128 waveform samples. In contrast, the original branch of the UMX model processes every 1024 waveform samples, as determined by the STFT hop size. We explain the design of the down-sampling block in the next paragraph. The down-sampled features are then converted from the dimension of 4800 to 512 by the linear layer and summed up with the original features from the UMX branch.

In the original UMX architecture, the input is the spectrogram of STFT. Since all modules in UMX work with the time resolution of STFT, the down-sampling block is required to align the time resolution between the SONIDO features and STFT. In the ablation study, we explored three different down-sampling operations: max pooling (MP), average pooling (AP), and unfolding (UF). Figure 11 illustrates the three different operations for the down-sampling block.

AP averages the multiple SONIDO features with the hop size and kernel size corresponding to the hop size and the window size of STFT, respectively. MP works similarly to AP but replaces the average pooling with a max pooling operation. In UF, the dimension of the SONIDO feature sequence is first converted from

Table 14: Evaluation results of music source separation on vocal extraction task

| Method | SDR [dB] | |
| | Vocals | Accompaniment |
| --- | --- | --- |
| Open-Unmix (UMX) | 5.71 | 11.57 |
| UMX with SONIDO (MP, EI) | 5.76 | 11.86 |
| UMX with SONIDO (AP, EI) | 5.55 | 11.71 |
| UMX with SONIDO (UF, EI) | **5.92** | **11.92** |
| UMX with SONIDO (UF, LI) | 5.83 | 11.68 |

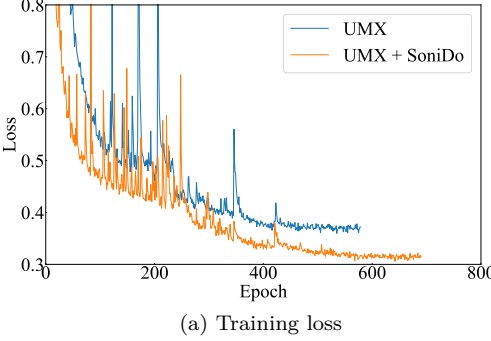

(a) Training loss

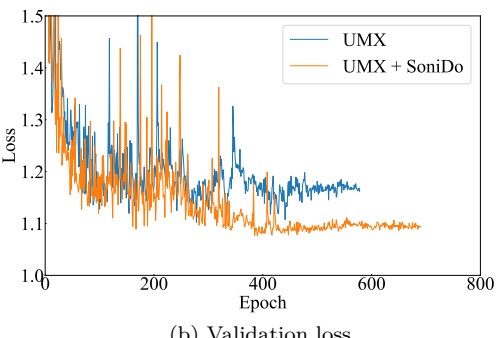

(b) Validation loss

Figure 12: Training and validation curves for UMX experiments in music separation task

4800 to 150 with a linear layer and ReLU function. Next, the feature sequence corresponding to the window size of STFT are concatenated every STFT hop size. After the concatenation, the dimension of the feature returns to 4800 again.

As well as the design of the down-sampling block, we further evaluated two methods for injecting the down-sampled features into UMX, as shown in Figure 10. With the "early injection" (EI) method, the SONIDO features are injected into UMX's encoder block. With the "late injection" (LI) method, the extracted features are injected into the decoder block.

We explored the optimal down-sampling and injection method for UMX on a vocal extraction task using MUSDB18. We followed the original training configuration of UMX for all experiments except that we disabled data augmentation for simplicity. To purely evaluate the trained components, MWF was skipped in all experiments.

### E.2 Results: Ablation study for UMX with SONIDO

The results of the aforementioned ablation study are summarized in Table 14.

For the down-sampling block, MP and AP did not improve the SDR score, whereas UF achieved a 0.34 dB improvement for vocals and 0.36 dB improvement for accompaniment. The results indicate that UF is the proper choice for the down-sampling block. We assume that the lower-level information is lost during the pooling operation over the temporal axis in MP and AP, while UF preserves such information by stacking multiple tokens into one frame in the unfolding manner.

Table 14 shows that EI is superior to LI. This suggests that it is the sequence modeling block in UMX that effectively used the SONIDO features to improve separation performance. The observation also inspired us to inject SONIDO features into the transformer block in HTDemucs.

Figure 12 shows the training and validation curves for UMX and UMX with SONIDO (UF, EI). The loss curve of UMX with SONIDO tends to be lower than that of UMX in both training and validation. *i.e.*, the number of epochs required to reach target loss was smaller when UMX was trained with SONIDO. The

converging loss of UMX with SoniDo was also lower than that of the original UMX, which reveals that the benefit from SoniDo can be consistently observed in the training phase, and the performance improvement is significant.

### E.3 Experiments: HTDemucs with SoniDo

We provide more details for music source separation with HTDemucs (Rouard et al., 2023) and SoniDo from Section 4.2.2. To investigate the effect of the SoniDo features, we trained several models on MUSDB18 (Rafii et al., 2017). We compare the following models:

- HTDemucs (default): Model with default settings (Dora[5] signature '955717e8'). The default batch size is 32 which corresponds to 4 samples per GPU as we trained in parallel on 8 GPUs. The default number of training epochs is 360.

- HTDemucs + SoniDo : Combination of HTDemucs with SoniDo. We introduced two new cross-domain transformer encoders into HTDemucs which are inserted directly after the original cross-domain transformer encoder. The first encoder facilitates information exchange between the spectral and SoniDo feature sequence, while the second one enables interaction between the waveform and SoniDo feature sequence. The SoniDo features are computed from the monaural downmix of the mixture using the top prior. Hence, the SoniDo feature sequence has a dimension of ($T = 2688$) × ($C = 4800$), reflecting the characteristics of training samples with a duration of 7.8 s. Subsequently, these features undergo normalization through a LayerNorm and further projected from their original dimension of 4800 to 2048 using a linear layer, giving them the same size as the features in the sequences from both the spectral and waveform branches of HTDemucs. Each additional cross-domain transformer encoder has a depth of 3, where we set `cross_first=True`. Due to the additional SoniDo model, we needed to reduce the batch size to 16 (corresponding to 2 samples per GPU as we trained on 8 GPUs). To keep the number of samples that the models seen during training the same, the number of training epochs was increased to 720. Additionally, to match the random remixing augmentation of the default HTDemucs model, we added 860 random mixes, generated from the 86 training songs in MUSDB18. It is important to note that this was done to ensure the same augmentation and does not introduce new songs to the training set; we still exclusively trained on the train split of MUSDB18.

- HTDemucs (ablation 1): Training with default settings '955717e8' where we also reduced the batch size to 16 and increased the number of training epochs to 720, together with the additional 860 random mixes used for HTDemucs + SoniDo .

- HTDemucs (ablation 2): Training as HTDemucs (ablation 1) but where the number of layers in the cross-domain transformer was increased from 5 to 11, matching the $2 \cdot 3 = 6$ additional transformer layers of HTDemucs + SoniDo .

- HTDemucs + STFT-2048: Same training settings as HTDemucs + SoniDo but where we used STFT features instead of SoniDo features. We computed the STFT with a Hann window of 2048 samples and hop size of 512 from the monaural downmix of the mixture.

- HTDemucs + STFT-4096: Same training settings as HTDemucs + SoniDo but where we used STFT features instead of SoniDo features. We computed the STFT with a Hann window of 4096 samples and hop size of 256 from the monaural downmix of the mixture.

- HTDemucs + CLAP: Same training settings as HTDemucs + SoniDo but where we used the embeddings from CLAP (Wu* et al., 2023) instead of the SoniDo features. More specifically, we used the 'fine-grained embeddings' before the final AP in CLAP. This experiment enabled us to compare features obtained with two trained models (as opposed to the STFT)[6].

---

[5]https://github.com/facebookresearch/dora
[6]Note that SoniDo and CLAP were trained on different datasets.

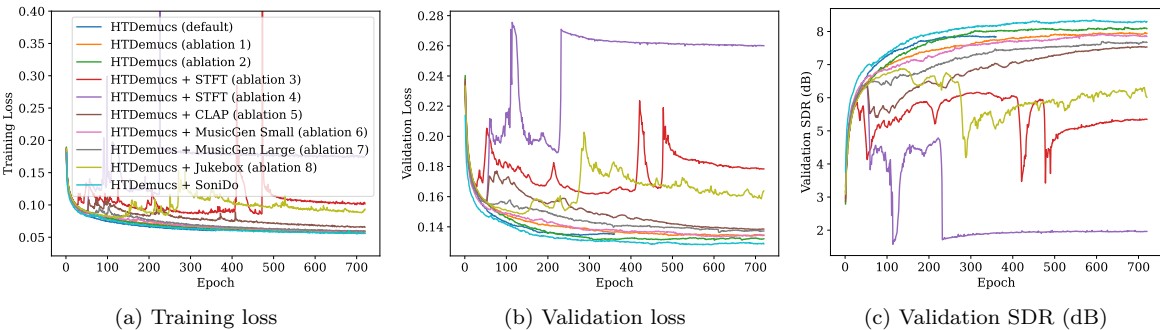

(a) Training loss          (b) Validation loss          (c) Validation SDR (dB)

Figure 13: Training and validation curves for HTDemucs experiments in music source separation task. We conditioned HTDemucs with four different features (STFT, CLAP, MusicGen and SONIDO). We achieved highest separation scores when injecting features of SONIDO. Interestingly, we observed instabilities when injecting STFT and CLAP features but not when injecting SONIDO features.

- **HTDemucs + MusicGen Small**: Same training settings as HTDemucs + SONIDO but where we used intermediate features from MusicGen (Copet et al., 2023) instead of the SONIDO features. More specifically, we used the 'musicgen-small' activations at the output of the 12th layer. This experiment enabled us to compare features obtained with two music foundation models[7].

- **HTDemucs + MusicGen Large**: Same training settings as HTDemucs + MusicGen but where we used the 'musicgen-large' activations at the output of the 24th layer. This enabled us to compare features obtained with two music foundation models with the same order of magnitude of number of learnable parameters. Note that, to reduce instability during training, this experiment used AdamW (Loshchilov & Hutter, 2017) as the optimizer, instead of Adam.

- **HTDemucs + Jukebox**: Same training settings as HTDemucs + SONIDO but where we used intermediate features from Jukebox (Dhariwal et al., 2020) instead of the SONIDO features. More specifically, we used the activations at the output of the 36th layer of the model '5b'[8].

Figure 13 displays the training and validation curves for these models.

## F Music Mixing

Music mixing is a crucial task in music production and typically conducted using audio processors or audio effects, which are signal processing systems that alter specific characteristics of the input signal. Several signal processing and machine learning methods have been investigated to automatize this task (Steinmetz et al., 2022), with the goal of simplifying the process for less experienced content creators and enhancing the workflow capabilities of professionals (Moffat & Sandler, 2019).

Data-driven deep learning approaches for automatic music mixing have focused on two fundamental frameworks: direct transformation networks, in which the model executes the mixing in a black-box manner, and parameter estimation networks, in which the mixing is carried out via differentiable audio processors. Martínez-Ramírez et al. (2021) proposed Mix-Wave-U-Net, a modified Wave-U-Net for drum mixing as a direct transformation system, while Steinmetz et al. (2021) introduced a differentiable mixing console in which neural proxies act as a parameter estimation network. Both systems have been acknowledged as having limited performance due to the scarcity of available training data, failing to meet professional audio engineering standards. Such systems require unprocessed or dry multitrack recordings and their corresponding mixtures, and large datasets are not readily accessible. To address this limitation, Martínez-Ramírez

---

[7]Note that SONIDO and MusicGen were trained on different datasets.
[8]Note that SONIDO and Jukebox were trained on different datasets.

et al. (2022) proposed an Fx-normalization preprocessing method that enables the training of direct transformation automatic mixing systems using processed or wet multi-track audio datasets, akin to the datasets used in source separation. Building on this approach, Koo et al. (2023) introduced a contrastive learning approach that allows a direct transformation network to execute mixing style transfer. Vanka et al. (2024) proposed a differentiable mixing style transfer system that predicts console parameters from raw tracks and a reference mix, advancing both parameter estimation and style transfer approaches.

While Koo et al. (2023) used SSL embeddings from a reference mixture to guide the mixing style transfer task, to the best of our knowledge, our approach is the first data-driven automatic mixing approach that incorporates SSL features from the input stems or high-level information related to genre, instrumentation, or mood to enhance automatic mixing performance.

### F.1 Details of Mix-Wave-U-Net and CRAFx2 with SONIDO

Mix-Wave-U-Net extends the U-Net architecture for audio signal processing tasks. We used FiLM layers (Perez et al., 2018) to incorporate the SONIDO features into each up-sampling 1D convolutional block and the bottleneck 1D convolutional block in the Mix-Wave-U-Net. Following Meseguer-Brocal & Peeters (2019), we positioned FiLM layers after the normalization layer and before the LeakyReLU activation function.

CRAFx2 (Martínez-Ramírez et al., 2022) comprises (1) an adaptive front-end that learns a filter bank, (2) latent-space mixer that learns a mixing mask functioning as equalizer, dynamic range compression, and reverberation transformations, and (3) a synthesis back-end that, through adaptive gains, implements loudness and panning transformations for each filter-bank channel. To enhance the mixing performance of the network, we also incorporated the SONIDO features into the relevant layers of the network. Following a similar approach to Mix-Wave-U-Net, we use FiLM layers to condition both the latent-space mixer and synthesis back-end. The latent-space mixer was first constructed on the basis of a temporal dilated convolutional network (TCN) followed by stacked bidirectional long short-term memory (BLSTM) layers, and FiLM layers were inserted within the TCN block and before the BLSTM layers. For the TCN part, the FiLM layers were placed after the depthwise convolution operation and before the second nonlinear activation function. Before the input of the BLSTM layers, we introduced a FiLM layer. Finally, the synthesis back-end incorporated a squeeze-and-excitation block (SE) (Hu et al., 2018), which scales channel-wise information by applying adaptive gains. A GroupNorm and FiLM layer are added after the second linear layer of the SE and before the sigmoid function.

### F.2 Experiments: Mix-Wave-U-Net and CRAFx2 with SONIDO

We used Mix-Wave-U-Net and CRAFx2 to explore the effect of the SONIDO features on the music mixing task. The input to all networks is the Fx-normalized (Martínez-Ramírez et al., 2022) stereo stems; vocals, bass, drums, and other, with the output being the stereo mixture. Each input stem consists of 2 channels of 10-s audio frames at 44.1 kHz. The SONIDO features are computed from the summation of the Fx-normalized input stems using the top prior and unconditional extraction. Therefore, the SONIDO feature sequence has a dimension of $(T = 3446) \times (C = 4800)$.

Although music mixing is a time-varying task, we hypothesize that high-level concepts such as genre, instrumentation, and mood, might improve mixing performance. Therefore, following the pre-processing for time-invariant retrieval described in Section 3.2, we carried out AP over the time dimension, and the tokens were time-averaged to a dimension of $(T = 1) \times (C = 4800)$, subsequently normalized via a non-trainable LayerNorm layer, and dimensionally reduced to 512 through a linear layer. We found that reducing the dimension too much (e.g., 128) decreased performance. The linear layer is followed by a dropout layer with a probability of 0.25 to avoid overfitting. All networks underwent the same token-processing steps.

From MUSDB18, 86 songs were used for training, and 14 and 50 for validation and testing, respectively. Since this dataset is significantly smaller than that used by Martínez-Ramírez et al. (2022), the number of batches per epoch was reduced from 1600 to 320, and all models were trained for 520 epochs. For the

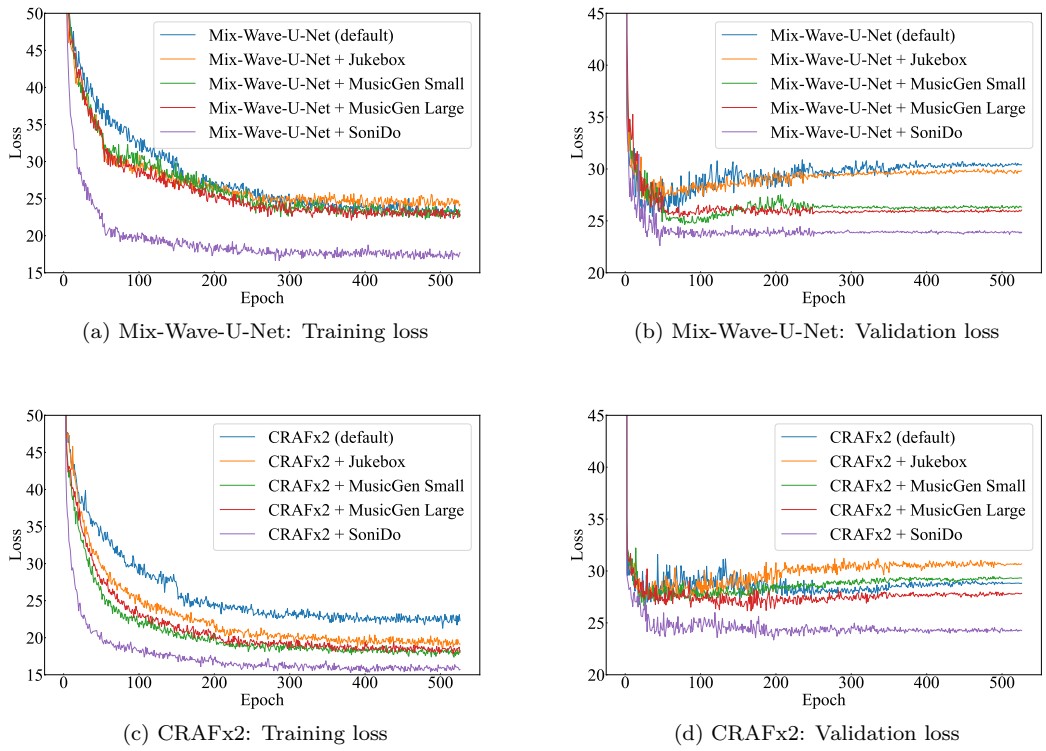

Figure 14: Training and validation curves for Mix-Wave-U-Net and CRAFx2 in music mixing task

default networks, we used the suggested initial learning rate of $1e^{-3}$. For models involving SoniDo, to ensure stability, we set the initial learning rate to $1e^{-4}$.

The loss function corresponds to the stereo-invariant loss that Martínez-Ramírez et al. (2022) reported as the best-performing, which they referred to as $L_b$, and consists of A-weighting pre-emphasis and low-pass finite impulse response filters, L2-norm on the spectral magnitude, and the L1-norm on the spectral log-magnitude.

To investigate the effect of the SoniDo features, we trained various models using features from both MusicGen Small and MusicGen Large. The training and feature injection settings are identical to those of Mix-Wave-U-Net + SoniDo and CRAFx2 +SoniDo, respectively, and the MusicGen features were extracted in the same manner as described in Appendix E.3/

To objectively evaluate the performance of all mixing systems, we used audio features related to the main audio characteristics that audio engineers manipulate during the mixing process, as shown in several works (Colonel & Reiss, 2021; Steinmetz et al., 2022; Martínez-Ramírez et al., 2022; Vanka et al., 2024). We computed the following audio features, spectral: centroid, bandwidth, contrast, roll-off, and flatness (Peeters, 2004); panning: the panning root mean square (PRMS) for total panning, low, mid, and high frequencies (Tzanetakis et al., 2007); dynamic: RMS level, dynamic spread, and crest factor (Ma et al., 2015); and loudness units full scale (LUFS) level and peak loudness (ITU-R, 2011). All features were computed using a running mean of 0.5-s (Tzanetakis et al., 2007). The objective evaluation test sets are identical to Martínez-Ramírez et al. (2022). These include MDXDB21-dry, an 18-song dry test set from MDXDB21, and the MUSDB18 test set, comprising 50 wet multi-track songs.

### F.3  Results: Mix-Wave-U-Net and CRAFx2 with SoniDo

From the training and validation curves in Figure 14, we can see that both Mix-Wave-U-Net + SoniDo and CRAFx2 + SoniDo exhibited a significant improvement during training and enhanced generalization

Table 15: Objective metrics correspond to mean absolute percentage error per low-level audio feature. Results are presented for MDXDB21-dry test set and MUSDB18 test set.

| Test set / Model | Spectral | | | | | Panning | | | | Dynamic | | | Loudness | |
|---|---|---|---|---|---|---|---|---|---|---|---|---|---|---|
| | centroid | band-width | contrast | roll-off | flatness | PRMS total | PRMS low | PRMS mid | PRMS high | RMS | spread | crest | LUFS | peak |
| *MDXDB21-dry* | | | | | | | | | | | | | | |
| Mix-Wave-U-Net (default) | 0.250 | 0.178 | 0.286 | 0.312 | 0.143 | 0.217 | 0.296 | 0.101 | 0.246 | 0.077 | 0.047 | 0.096 | 0.094 | 0.243 |
| Mix-Wave-U-Net + Jukebox | 0.236 | 0.176 | 0.300 | 0.322 | **0.140** | 0.278 | 0.236 | 0.084 | 0.325 | 0.078 | 0.047 | 0.100 | 0.086 | 0.222 |
| Mix-Wave-U-Net + MusicGen Small | 0.265 | 0.182 | **0.273** | 0.324 | 0.147 | 0.192 | 0.266 | 0.119 | 0.211 | **0.065** | 0.044 | **0.082** | 0.087 | 0.206 |
| Mix-Wave-U-Net + MusicGen Large | 0.264 | 0.191 | 0.286 | 0.323 | 0.149 | 0.262 | **0.243** | 0.130 | 0.288 | **0.065** | 0.044 | 0.089 | **0.083** | 0.207 |
| Mix-Wave-U-Net + SoniDo | **0.228** | **0.159** | 0.295 | **0.290** | 0.158 | **0.183** | 0.243 | **0.088** | **0.205** | 0.071 | **0.041** | 0.089 | 0.088 | **0.173** |
| | | | | | | | | | | | | | | |
| CRAFx2 (default) | **0.216** | 0.165 | 0.196 | **0.264** | **0.123** | 0.168 | **0.273** | **0.078** | 0.197 | 0.072 | 0.049 | 0.087 | 0.086 | 0.218 |
| CRAFx2 + Jukebox | 0.282 | 0.206 | 0.176 | 0.339 | 0.151 | 0.266 | 0.333 | 0.082 | 0.315 | 0.082 | 0.046 | 0.098 | 0.084 | 0.204 |
| CRAFx2 + MusicGen Small | 0.262 | 0.195 | 0.172 | 0.338 | 0.144 | 0.265 | 0.312 | 0.079 | 0.318 | 0.081 | 0.047 | 0.089 | 0.090 | 0.209 |
| CRAFx2 + MusicGen Large | 0.274 | 0.215 | **0.170** | 0.334 | 0.147 | 0.230 | 0.292 | 0.086 | 0.267 | 0.077 | 0.047 | 0.094 | **0.075** | **0.190** |
| CRAFx2 + SoniDo | 0.226 | **0.157** | 0.273 | 0.283 | 0.169 | **0.145** | 0.307 | 0.085 | **0.162** | **0.068** | **0.044** | **0.084** | 0.109 | 0.232 |
| *MUSDB18* | | | | | | | | | | | | | | |
| Mix-Wave-U-Net (default) | 0.260 | 0.173 | 0.170 | 0.291 | 0.109 | 0.156 | 0.222 | 0.104 | 0.172 | 0.087 | 0.069 | 0.098 | 0.094 | 0.241 |
| Mix-Wave-U-Net + Jukebox | 0.273 | 0.179 | 0.169 | 0.302 | 0.108 | 0.211 | 0.204 | 0.094 | 0.241 | 0.089 | 0.059 | 0.098 | **0.089** | **0.225** |
| Mix-Wave-U-Net + MusicGen Small | 0.294 | 0.192 | **0.162** | 0.301 | 0.119 | **0.145** | **0.211** | 0.116 | **0.160** | 0.088 | 0.056 | 0.093 | 0.097 | 0.230 |
| Mix-Wave-U-Net + MusicGen Large | 0.276 | 0.176 | 0.170 | 0.299 | **0.107** | 0.203 | 0.232 | 0.108 | 0.225 | 0.079 | 0.057 | 0.090 | 0.096 | 0.237 |
| Mix-Wave-U-Net + SoniDo | **0.240** | **0.152** | **0.162** | **0.265** | 0.110 | 0.179 | 0.217 | **0.103** | 0.200 | **0.068** | **0.050** | **0.069** | **0.089** | 0.269 |
| | | | | | | | | | | | | | | |
| CRAFx2 (default) | 0.253 | 0.173 | 0.152 | **0.274** | **0.111** | **0.132** | 0.253 | **0.086** | 0.147 | 0.097 | 0.047 | 0.098 | **0.089** | 0.241 |
| CRAFx2 + Jukebox | 0.284 | 0.210 | 0.154 | 0.303 | 0.129 | 0.238 | 0.264 | 0.104 | 0.275 | 0.097 | 0.047 | 0.103 | **0.089** | 0.241 |
| CRAFx2 + MusicGen Small | 0.279 | 0.190 | 0.157 | 0.303 | 0.125 | 0.223 | **0.242** | 0.091 | 0.258 | 0.098 | 0.047 | 0.105 | 0.101 | 0.255 |
| CRAFx2 + MusicGen Large | 0.293 | 0.212 | 0.163 | 0.317 | 0.136 | 0.209 | 0.277 | 0.099 | 0.237 | **0.093** | 0.048 | 0.098 | 0.099 | 0.251 |
| CRAFx2 + SoniDo | **0.243** | **0.157** | **0.148** | 0.276 | 0.113 | **0.132** | 0.249 | 0.088 | **0.146** | **0.093** | **0.046** | **0.089** | 0.097 | **0.240** |

during validation. These results are reflected in Table 6, where the models that included SoniDo tokens consistently displayed improved performance of the stereo-invariant loss. Regarding the audio effect-related features, Table 15 shows that across most feature categories, Mix-Wave-U-Net + SoniDo outperformed the default and Mix-Wave-U-Net.

Incorporating Jukebox and MusicGen features generally leads to improvement over the default network, although not as much as SoniDo. MusicGen Small performed slightly better than the large model, confirming the trend observed in previous downstream tasks. However, for both MusicGen and Jukebox, improvement over the default network is not always the case with CRAFx2. In the MDXDB21-dry test set, the default model performed better than MusicGen in terms of spectral and panning features as well as stereo-invariant loss. Yet, MusicGen did enhance the dynamic and loudness features, while Jukebox only improved loudness and generally led to overfitting. The performance gap of MusicGen in spectral-related metrics, and generally when compared with SoniDo, might stem from MusicGen being trained on 32-kHz audio, while both our evaluation datasets and training dataset of SoniDo correspond to 44.1-kHz audio.

This discrepancy in training data sampling rates leads us to assume that the relatively lower performance of MusicGen may be attributed to this difference. Further investigation is required to confirm this hypothesis and explore the impact of sampling rates on music foundation models as boosters for music mixing tasks.

It is worth noting that the inherent differences between the MDXDB21-dry and MUSDB18 test sets are reflected in these scores. This is expected, given that the MDXDB21-dry data consists of dry multi-tracks, presenting a more realistic mixing scenario than the wet multi-tracks of MUSDB18. Therefore, we attribute the substantial difference in reported stereo-invariant values in Table 6 to the fact that the target mixture has been processed with different audio processors, and the timbral characteristics cannot always be matched by the trained networks.

The mismatch in performance between the stereo-invariant loss and low-level features related to audio effects reflects the challenge of objectively evaluating automatic mixing systems, which remains an ongoing and open research direction (Stables et al., 2019; Steinmetz et al., 2022). Thus, for a more in-depth analysis of such systems, a listening test is required. However, for the scope of this paper, the reported objective results support our hypothesis that incorporating the SoniDo features, i.e. high-level knowledge of the input stems to be mixed, overall improves training and generalization when these features are properly used.

