# OpenReview forum: "Music Foundation Model as Generic Booster for Music Downstream Tasks"
_TMLR — Accepted by TMLR_

### Review · Reviewer_Gxn5 · 2025-01-27

**Summary Of Contributions:**

SoniDo employs a unique approach compared to  other foundational music models such as Jukebox, MusicLM, and MusicGen. Jukebox utilizes multi-resolution VQ-VAEs, where input audio for each resolution is encoded and decoded separately. MusicLM and MusicGen rely on RVQ (Residual Vector Quantization) within an encoded space, followed by decoding. SoniDo uses a HQ-VAE architecture, which functions similarly to Jukebox but introduces a hierarchical model. Instead of having a single encoder for each resolution, SoniDo creates different encoded spaces for each resolution that are jointly decoded. This hierarchical encoding and decoding mechanism sets it apart from the others.

Another distinguishing feature of SoniDo is its custom loss function. First, there is an L2 reconstruction loss based on the input and output. Next, for each hierarchical layer, there is an L2 loss on the before and after quantized features followed by the subtraction of the entropy of the pmf. The loss is also regularized with respect to the variance of p(x|Z).

In the second stage, SoniDo trains three distinct autoregressive sparse transformers to model priors. SoniDo introduces a new data augmentation technique called token-out, which prevents overfitting. Token-out takes the concatenation of the three hierarchical layers’ features and randomly masks tokens, followed by a shuffling.

SoniDo has demonstrated exceptional performance in various experiments. SoniDo has shown that the features extracted used for music understanding can be used for downstream tagging tasks. SoniDo passes state of the art for many tasks. For music transcription, it has been shown to significantly enhance transcription accuracy. In source separation tasks, combining SoniDo with HTDemucs consistently delivers superior results compared to other studies. For music mixing, adding SoniDO to existing models has been shown to reach state of the art for the majority of tasks.

**Audience:**

Yes

**Broader Impact Concerns:**

All ethical concerns, including licensing and bias in the model, have been addressed in the ethics section and introduction.

**Claims And Evidence:**

Yes

**Requested Changes:**

Strengthen Work
- Change position of diagram captions so that they are all the same position relative to the diagram
- Add some phrasing in the last paragraph of 3.2 to talk about the masking over the whole sequence rather than masking each layer independently.

Critical Changes
- No major changes

**Strengths And Weaknesses:**

Strong Elements
- Goal of the paper is to increase performance by using hierarchical intermediate features and to verify that extracted features contain good information on music understanding.
- - This paper achieves this goal by implementing a HQ-VAE specifically for music audio.
- - This paper solidifies claims by running many downstream tasks and comparing to the SOTA in each respective task
- Specifically lists out the SOTA for each downstream task

Weak Elements
- Is token out applied to the whole sequence, or for each hierarchical layer (Figure 4)?
Small Weak Elements
- Position of diagram captions are not consistently on top or on bottom of diagram

---

> ### Author Response · Authors · 2025-03-23
> **Author reply**
>
> Thank you for your constructive feedback and recognition of our work. We would like to address your concerns below.
>
> **Requested Changes**
>
> > (RC1) Change position of diagram captions so that they are all the same position relative to the diagram
>
> We have modified the manuscript so that the captions for Tables 14 and 15 are now positioned above the tables.
>
> ---
>
> > (RC2) Add some phrasing in the last paragraph of 3.2 to talk about the masking over the whole sequence rather than masking each layer independently.
>
> We have revised the last paragraph of Section 3.2 to emphasize that the masking procedure is applied to the entire token sequence.

---

### Review · Reviewer_ukVv · 2025-02-28

**Summary Of Contributions:**

The paper proposes “SoniDo,” a new foundation model for audio tasks and thoroughly characterizes its audio encoding capabilities over various downstream music understanding and mixing tasks. Fundamentally, SoniDo differs from other foundation models through its underlying quantization setup (HQ-VAE vs. VQ-VAE / RVQ) which promises better encoding capabilities. SoniDo embeddings seem to boost downstream performance when introduced in various task-specific specialized models compared to using MusicGen embeddings.

**Audience:**

Yes

**Broader Impact Concerns:**

While the authors do address ethical concerns about their dataset, more details on how the dataset was procured, what kind of data does the dataset have, etc. should greatly improve the ethical aspect of the paper.

**Claims And Evidence:**

Yes

**Requested Changes:**

Below are some follow-up questions on the paper:

- MusicGen small seems to always perform better than large? See tables 2, 3, 4, 5, and 6. Is this expected?
- Since the training dataset for the foundation models is different, how to attribute performance delta to the proposed change of HQ-VAE vs. VQ-VAE / RVQ?
- What is the open-sourcing plan of SoniDo? Are there plans to open-source the dataset, model weights, or anything else?

**Strengths And Weaknesses:**

Positives:
- First application of the HQ-VAE parameterization in music foundation models
- State of the art performance for various downstream tasks

Negatives:
- Despite SoniDo being a generative model, there are very limited results on generative audio tasks (only music mixing seems to be covered), e.g., text to audio, audio to audio, etc.

---

> ### Author Response · Authors · 2025-03-24
> **Author reply**
>
> We greatly appreciate the valuable feedback you have provided, and have addressed each of your comments. Please kindly find our response below.
>
>
> **Requested Changes**
>
> > (RC1) MusicGen small seems to always perform better than large? … Is this expected?
>
> The method used to extract knowledge from a pretrained model significantly influences downstream task performance, which can be categorized into generation tasks and understanding tasks. However, the relationship between the performance of the pretraining task (i.e., a pure generation task), which generally correlates positively with model size but may be influenced by other factors such as latent structure and probing method, and the performance on downstream tasks is not straightforward. Therefore, it is possible that MusicGen-small may outperform its larger counterpart, a finding that aligns with the results in the original paper [1] and concurrent works [2, 3].
>
> [1] Copet et al, “Simple and Controllable Music Generation,” NeurIPS (2023).
>
> [2] Ma et al., “Do music LLMs learn symbolic concepts? A pilot study using probing and intervention”, NeurIPS Workshop Audio Imagination (2024).
>
> [3] Wei et al., “Do Music Generation Models Encode Music Theory?” ISMIR (2024).
>
> ---
>
> > (RC2) Since the training dataset for the foundation models is different, how to attribute performance delta to the proposed change of HQ-VAE vs. VQ-VAE / RVQ?
>
> The primary claim of our paper is that using features extracted from the music foundation model enhances the performance of downstream task models. We employ a hierarchical model that allows us to adapt the granularity of representations for different downstream tasks by selecting the appropriate layers. Our empirical results confirm that the choice of layers significantly impacts task performance. We acknowledge that performing a fair comparison to isolate the effects of HQ-VAE versus VQ-VAE/RVQ is challenging and plan to address this in future work.
>
> ---
>
> > (RC3) What is the open-sourcing plan of SoniDo?
>
> We understand that open-sourcing will contribute greatly to the research community. However, although our contract for the internal dataset allowed research exploration using the dataset, it is not compatible for open-sourcing the resulting model. As a result, we may not open-source SoniDo, but hope that the methodology described in the work can be helpful for future research.
>
>
> **Additional response**
>
> > more details on how the dataset was procured
>
> We obtained the permit to use the dataset through a research-specific contract. All the tracks in the dataset were composed, recorded, mixed and mastered by its provider.
>
> ---
>
> > there are very limited results on generative audio tasks
>
> Although our foundation model is designed with generative modeling in mind, the focus of this paper is on leveraging its features to improve various music-related downstream tasks. Nonetheless, we evaluate our model's generative capabilities in Appendix B.3. Notably, unlike the other baselines presented in Table 8, our model generates music signals at 44.1 kHz, demonstrating its suitability for downstream tasks requiring studio-quality audio.

---

### Review · Reviewer_8o1p · 2025-03-10

**Summary Of Contributions:**

This paper presents a methodology in which intermediate features from a Music Foundation Model are used to enhance various downstream music tasks, such as music tagging, music transcription, source separation, and music mixing, resulting in performance improvements in some cases.

**Audience:**

Yes

**Broader Impact Concerns:**

Nothing in particular besides the points mentioned in the Ethical Concerns section.

**Claims And Evidence:**

Yes

**Requested Changes:**

To strengthen the work, the main proposed change would be to include more baselines than just MusicGen for the Music Foundation Model, in order to better contextualize the work and demonstrate the effectiveness of the extracted intermediate features. MFMs do not necessarily need to be restricted to the tokenized audio + language modeling setting. (For a variety of examples, see: Ma, Yinghao, et al. 'Foundation models for music: A survey.' arXiv preprint arXiv:2408.14340 (2024).) Also, providing more details— even at a high level— and statistics regarding the dataset would improve the understanding of the affordances of the intermediate representations, particularly in terms of any potential biases. Moreover, clarifying the definition of Music Foundation Models would enhance the paper contextually.

**Strengths And Weaknesses:**

Strengths:
The paper is clearly written, and the architecture is described in detail. The suite of downstream music tasks is comprehensive. Pushing the frontier of using music foundation models for powerful representations and downstream music tasks is valuable for the field. In some settings, SOTA scores have been improved.

Weaknesses:
The main weakness of this paper is that it lacks baselines for its primary claim of using extracted features from a music foundation model for downstream music tasks. The paper experimentally compares the presented MFM only against MusicGen in the experiments section (for music transcription, music source separation and music mixing). Music foundation models should not necessarily be restricted to the tokenized audio + language modeling ecosystem, and even within this particular setting, benchmarking the presented MFM solely against MusicGen doesn't provide enough context to demonstrate the full potential of the extracted intermediate representations. It mainly supports the claim that using feature extraction for certain downstream tasks can lead to performance improvements. As it stands, it's difficult to contextualize the presented methodology.

The lack of dataset availability makes it more challenging to interpret the quality of intermediate features, particularly in light of potential biases.

Additionally, in the context of this paper, the definition of a Music Foundation Model is unclear (any representation learning setting?, multi-task adaptability?, music tagging, music transcription, music source separation, music mixing and beyond?).

---

> ### Author Response · Authors · 2025-03-24
> **Author reply**
>
> Thank you for reviewing our paper carefully. Your comments and questions are valuable for improving our work. We have thoroughly considered each of your points and have provided our response below.
>
>
> **Requested Changes**
>
> > (RC1) the main proposed change would be to include more baselines than just MusicGen for the Music Foundation Model
>
> We appreciate your suggestion to refer to the survey paper [1], which serves as a valuable reference to contextualize our work. We agree that Music Foundation Models are not limited to generative modeling with tokenized audio. However, we note that (a) most models in other directions—such as contrastive learning and masked modeling—mainly focus on understanding tasks such as tagging (see Table I in [1]), or (b) they tend to perform significantly worse than task-agnostic models on more complex tasks like source separation (see Figure 4 in [1]). Our empirical results partially support (b): Table 1 in our paper shows that SoniDo and MusicGen, both generative modeling approaches, significantly outperform MERT, a masked language modeling approach, in source separation tasks. Since our study covers not only understanding tasks but also generative tasks such as source separation and music mixing, we opted to compare our model with representative generative models.
>
> Since Jukebox is the most relevant foundation model to SoniDo, and it’s still performing reasonably well in some downstream tasks according to some recent works [2, 3],  we are currently working on experiments to include it as a baseline for **all downstream tasks**. This will complete the “CALM w/ Jukebox-5B” column in Table 1. We kindly ask for your patience as we finalize these new results.
>
> ---
>
> > (RC2) more details— even at a high level— and statistics regarding the dataset
>
> Our internal dataset contains around 115k studio-quality library music tracks sampled at 44.1 kHz. Their lengths vary from 30s to 150s, their tempos vary between 50bpm to 200bpm, and the total length is around 4,000h. 90% of the dataset is non-vocal. Although there are more than 50 genres included in the dataset, it is biased toward orchestral and western music. We will include these high-level information into Appendix C to enhance the knowledge about the dataset for readers.
>
> ---
>
> > (RC3) clarifying the definition of Music Foundation Models would enhance the paper contextually
>
> We define a Music Foundation Model as a pre-trained model developed on a large-scale dataset that is adaptable to various music-related downstream tasks, in line with the definition provided in [1]. In particular, we focus on auto-regressive models as they’re the direct analogy  when compared with foundation models defined in the NLP research field, which correspond to the APC design mentioned at section 4.2.c in [1]. In the context of music foundation models, we categorize downstream tasks into two types: understanding tasks and generative tasks. Accordingly, we have selected two representative tasks for each category—tagging and transcription for understanding tasks, and mixing and source separation for generative tasks—to cover the spectrum. Evaluating our music foundation model on additional tasks is a potential direction for future research.
>
>
> **Additional response**
>
> > it's difficult to contextualize the presented methodology
>
> In line with our response to (RC1) and (RC3), we are revising the introduction to better convey the position of our work during the discussion period.
>
> [1] Ma et al, “Foundation Models for Music: A Survey,” arXiv preprint, 2024.
>
> [2] Koo et al, "Understanding and Controlling Generative Music Transformers by Probing Individual Attention Heads", IEEE ICASSP Satellite Workshop on Explainable Machine Learning for Speech and Audio (XAI-SA), April 2024.
>
> [3] Wei et al., “Do Music Generation Models Encode Music Theory?” ISMIR (2024).

---

> > ### Author Response · Authors · 2025-03-24
> > **Follow-up reply**
> >
> > We have revised the introduction to better convey the context of our work.

---

> > > ### Author Response · Authors · 2025-04-11
> > > **Further update on experiments**
> > >
> > > Thank you for your patience. We have conducted the experiments as promised in our initial response and have updated the relevant tables (Tables 1, 3, 4, 5, 6, 13, and 15) in the manuscript accordingly. While Jukebox is known for its strong performance in understanding tasks, SoniDo is competitive with Jukebox in these tasks and outperforms it in generative tasks, which aligns with our expectations.

---

### Author Response · Authors · 2025-04-11
**To the AC and all reviewers**

First of all, we would like to express our sincere appreciation to all the reviewers for dedicating their time and expertise to reviewing our manuscript once again. In response to the reviewers' comments, we have added more experiments to our manuscript and have carefully updated it, with changes highlighted in red. Thank you for your patience as we finalized the experiments. The summary of our updates is as follows:

1. [**8o1p**] We tested Jukebox on all downstream tasks (Tables 3, 4, 5, 6, 13, and 15) and completed the "Jukebox-5B" column in Table 1.
2. [**8o1p**] We included high-level information about our internal dataset in Appendix B.
3. [**8o1p**] We modified the introduction to clarify our definition of a music foundation model and the position of this work.
4. [**Gxn5**] We fixed the placement of the diagram captions for Tables 14 and 15.
5. [**Gxn5**] We revised the last paragraph of Section 3.2 to clarify our masking procedure to avoid overfitting.
6. [**8o1p**, **ukVv**, **Gxn5**] We answered all the questions raised by the reviewers in our response.

---

### Decision · Action_Editor_qqw5 · 2025-04-23

**Recommendation:** Accept with minor revision

**Comment:**

One of the reviewers had a recommendation that was leaning reject but the recommendation was based on novelty, which is not a key criterion for TMLR acceptance.  Another reviewer (8o1p) leaned accept but noted various things in their recommendation.  I am pasting these below:

"
RC1) "most models in other directions—such as contrastive learning and masked modeling—mainly focus on understanding tasks such as tagging" -> It doesn't necessarily mean that models focusing on music understanding do not learn useful representations for many downstream tasks including generative, for example, we have seen similar success with CLIP + BigGAN back then.

"they tend to perform significantly worse than task-agnostic models on more complex tasks like source separation". Besides source separation, in Figure 4, what authors suggest here would be a too-quick-conclusion, as there is no strong evidence for that in the Figure.

RC2) Thanks for more information.

RC3) The definition of a Music Foundation Model proposed in this work is reasonable; however, the current study does not fully meet the criteria to warrant that designation. To qualify as a music foundation model, (a) the model should arguably be trained on a large and diverse music dataset. While the training set used in this study appears to be of adequate size relative to what's available in the literature, its musical diversity and coverage is highly questionable. Additionally, (b) the model and its learned representations should demonstrate broad utility across a wide range of downstream musical applications. In this case, the downstream tasks explored are not comprehensive enough—particularly lacking in generative music scenarios such as conditioning, inpainting, and style transfer to name a few, across various modeling paradigms.

My main point is that the context of this work remains difficult to justify. While I believe this research is a valuable contribution and deserves to be shared with the community, referring to it as a "Music Foundation Model" overstates its scope. Although the authors have provided additional information regarding the dataset, there is still no clear evidence that the learned representations support general music understanding. Moreover, given the limited range of downstream musical tasks evaluated, it is difficult to argue that this is a general-purpose, one-for-many foundation model.

What the study demonstrates is that SoniDo performs well on several downstream tasks—such as music tagging, transcription, source separation, and mixing—and, in some cases, surpasses models like MusicGEN, as well as Jukebox and MERT with more limited experimental settings. However, it does not warrant the broader label of a Music Foundation Model. My primary concern with the paper lies in its contextualization and rhetoric, which could be improved to make more balanced claims and better position the work.
"

It seems the reviewer's biggest concern is the presentation of the paper more than anything.  I think this can be addressed by appropriately bringing the paper into context, and perhaps toning done some of the language, as recommended above by the reviewer.

**Audience:**

Yes, I for one find the results interesting.  And there is a decently-sized community of researchers within the broader TMLR community working in AI for music, who would be interested in this paper.

**Claims And Evidence:**

For the most part, yes.  One reviewer noted in their recommendation that "this paper backs up its claims through the many in-depth studies in the appendix section. For each downstream task, there are many ablation studies."  Another reviewer (see comments below) continued to have some questions about the claims post-rebuttal.  I think these can be addressed in a minor revision.